# Powerful Potential of Polyfluoroalkyl-Containing 4-Arylhydrazinylidenepyrazol-3-ones for Pharmaceuticals

**DOI:** 10.3390/molecules28010059

**Published:** 2022-12-21

**Authors:** Yanina V. Burgart, Natalia A. Elkina, Evgeny V. Shchegolkov, Olga P. Krasnykh, Galina F. Makhaeva, Galina A. Triandafilova, Sergey Yu. Solodnikov, Natalia P. Boltneva, Elena V. Rudakova, Nadezhda V. Kovaleva, Olga G. Serebryakova, Mariya V. Ulitko, Sophia S. Borisevich, Natalia A. Gerasimova, Natalia P. Evstigneeva, Sergey A. Kozlov, Yuliya V. Korolkova, Artem S. Minin, Anna V. Belousova, Evgenii S. Mozhaitsev, Artem M. Klabukov, Victor I. Saloutin

**Affiliations:** 1Postovsky Institute of Organic Synthesis of the Ural Branch of the Russian Academy of Science (IOS UB RAS), S. Kovalevskoi St., 22, 620108 Ekaterinburg, Russia; 2Scientific and Educational Center for Applied Chemical-Biological Research, Perm National Research Poly-Technic University, Komsomolsky Av., 29, 614990 Perm, Russia; 3Institute of Physiologically Active Compounds at Federal Research Center of Problems of Chemical Physics and Medicinal Chemistry, Russian Academy of Sciences (IPAC RAS), Severny proezd 1, 142432 Chernogolovka, Russia; 4Institute of Natural Sciences and Mathematics, The Ural Federal University Named after the First President of Russia B. N. Yeltsin, Lenina Av., 51, 620083 Ekaterinburg, Russia; 5Ural Research Institute for Dermatology, Venereology and Immunopathology, Shcherbakova St., 8, 620076 Ekaterinburg, Russia; 6Shemyakin-Ovchinnikov Institute of Bioorganic Chemistry RAS, Miklukho-Maklaya St., 16/10, 117997 Moscow, Russia; 7M.N. Mikheev Institute of Metal Physics of the Ural Branch of the Russian Academy of Sciences, S. Kovalevskoi 18, 620108 Ekaterinburg, Russia; 8Institute of Immunology and Physiology of the Ural Branch of the Russian Academy of Sciences, Pervomayskaya 106, 620108 Ekaterinburg, Russia; 9N.N. Vorozhtsov Novosibirsk Institute of Organic Chemistry of Siberian Branch of Russian Academy of Sciences, 9 Lavrentiev Avenue, 630090 Novosibirsk, Russia; 10Smorodintsev Research Institute of Influenza of the Ministry of Health of the Russian Federation, 15/17 prof. Popov Street, 197376 Saint-Petersburg, Russia

**Keywords:** 4-arylhydrazinylidenepyrazol-3-ones, ADME profile, carboxylesterase inhibitor, antioxidant activity, cytotoxicity, antimicrobial activity, antinociceptive effect, TRPV1 receptor inhibitor

## Abstract

4-Arylhydrazinylidene-5-(polyfluoroalkyl)pyrazol-3-ones (4-AHPs) were found to be obtained by the regiospecific cyclization of 2-arylhydrazinylidene-3-(polyfluoroalkyl)-3-oxoesters with hydrazines, by the azo coupling of 4-nonsubstituted pyrazol-5-oles with aryldiazonium chlorides or by the firstly discovered acid-promoted self-condensation of 2-arylhydrazinylidene-3-oxoesters. All the 4-AHPs had an acceptable ADME profile. Varying the substituents in 4-AHPs promoted the switching or combining of their biological activity. The polyfluoroalkyl residue in 4-AHPs led to the appearance of an anticarboxylesterase action in the micromolar range. An NH-fragment and/or methyl group instead of the polyfluoroalkyl one in the 4-AHPs promoted antioxidant properties in the ABTS, FRAP and ORAC tests, as well as anti-cancer activity against HeLa that was at the Doxorubicin level coupled with lower cytotoxicity against normal human fibroblasts. Some Ph-N-substituted 4-AHPs could inhibit the growth of *N. gonorrhoeae* bacteria at MIC 0.9 μg/mL. The possibility of using 4-AHPs for cell visualization was shown. Most of the 4-AHPs exhibited a pronounced analgesic effect in a hot plate test in vivo at and above the diclofenac and metamizole levels except for the ones with two chlorine atoms in the aryl group. The methylsulfonyl residue was proved to raise the anti-inflammatory effect also. A mechanism of the antinociceptive action of the 4-AHPs through blocking the TRPV1 receptor was proposed and confirmed using in vitro experiment and molecular docking.

## 1. Introduction

The pyrazolone core is a favorable frame for the rational design of small molecule drugs [1,2,3,4,5,6]. Interest in this pharmacophore moiety arose immediately after the discovery of the analgesic and antipyretic activities of antipyrine (1,2-dihydro-1,5-dimethyl-2-phenylpyrazol-3-one). Based on it, a large number of pyrazolone analgesics–antipyretics (aminopyrine, propyphenazone, metamizole, nifenazone, famprofazone, morazone, etc.) have been developed [7]. This series is growing stably due to the synthesis of new derivatives [8], while the synthetic precursor of most drugs, named Edarovone™ (https://go.drugbank.com/drugs/DB12243 (accessed on 13 December 2022)), is used as a medication for stroke recovery, to treat amyotrophic lateral sclerosis and as a free radical scavenger, nootropic and neuroprotective drug [9].

The functionalization of pyrazolones by the (het)arylhydrazone moiety at position 4 is one of the directions for developing promising new bioactive compounds. A lot of investigations have been devoted to determining the activity of 5-substituted 4-arylhydrazinylidene-1,2-dihydropyrazol-3-ones as antimicrobial, antibacterial or antifungal agents [10,11,12,13,14] and anticancer remedies [15,16,17,18,19]. This backbone was used for the design of antiviral agents [20], including HIV-1 inhibitors [21,22,23,24], potent inhibitors of glycogen synthase kinase-3β [25], nonpeptidyl promoters of megakaryocytopoiesis with thrombopoietic activity [26], inhibitors of human amyloid polypeptide aggregation [27], human carbonic anhydrase and acetylcholinesterase [28], type II diabetes-related enzyme inhibitors [29], analgesic and anti-inflammatory drugs [11,13,30,31] and antioxidant agents [12,32].

The most significant can be considered the creation of eltrombopag olamine [3-(N-[1-(3,4-dimethylphenyl)-3-methyl-5-oxo-1,5-dihydropyrazol-4-ylidene]hydrazine)-2-hydroxybiphenyl-3-carboxylic acid], which is a nonpeptide thrombopoietin receptor agonist that has been approved by the FDA for the treatment of chronicimmune thrombocytopenia [33,34,35]. In addition to the medicinal chemistry, the 4-arylhydrazinylidene-2,4-dihydropyrazol-3-one (4-AHP) core was introduced in the industrial production of food colorings and dyes for textiles and other materials [36,37,38,39,40]. The antioxidant [41] and antimicrobial properties of 4-AHPs have opened new possibilities for their use as edible dyes [42].

The bioactivity of polyfluoroalkyl-containing 4-AHPs is poorly understood, although the introduction of fluorine atoms or a trifluoromethyl group into the organic molecule is one of the most popular routes for new drugs development [43,44,45,46,47]. Derivatives of 5-trifluoromethyl-4-AHP nucleosides were published as having anticancer [48,49] and antiviral [50] activity. It is known that their N- and N,O-bis-β-D-glycosides exhibit antimicrobial activity [51]. 4-AHPs having the second N-pyrazole moiety also show antimicrobial properties [52], while ones containing N-naphthyl di- and tri-sulfonic acids inhibit HIV-1 infection [21]. However, the biological potential of 4-AHPs has not been fully explored.

Two approaches are widely used for the synthesis of 4-AHPs: one of them is based on the cyclization of 2-arylhydrazinylidene-3-oxoesters with different α-N,N-dinucleophiles (hydrazines or hydrazides) [24,53] and the second method relies on azo-coupling pyrazolones with diazonium salts [54,55,56,57,58]. In addition, the step-economic iodine-mediated construction of functionalized 4-AHPs in the presence of catalytic AgNO_3_ starting from 3-oxoesters and two equivalents of arylhydrazines was proposed [59]. The one-pot silver trifluoromethanesulfonate catalyzed cascade reaction of 2-diazo-3-oxoesters with two arylhydrazines or two arylhydrazine hydrochlorides was described [60]. The reactions of 4-acetylsydnones with arylhydrazines were found to result in 4-AHPs [61].

There are few examples of the synthesis of polyfluoroalkyl-containing 4-AHPs. 5-Trifluoromethyl-4-AHPs were obtained *via* the azo coupling of 5-trifluoromethylpyrazol-3-ones with aryldiazonium chlorides [51], while 5-nonafluorobutyl-substituted 4-[2-(4-methoxyphenyl)hydrazinylidene]-2,4-dihydro-3*H*-pyrazol-3-one was prepared by the cyclization of the corresponding 2-arylhydrazinylidene-3-oxoester with hydrazine hydrate [62,63]. 2-Phenyl-4-(2-phenylhydrazinylidene)-5-(trifluoromethyl)-2,4-dihydro-3*H*-pyrazol-3-one was synthesized by the iodine-mediated reaction of ethyl trifluoroacetoacetate with two equivalents of phenylhydrazine in the presence of catalytic AgNO_3_ [59].

Herein, we investigated the bioactivity potential of polyfluoroalkyl-containing 4-AHPs, varying their structural elements: a polyfluoroalkyl group, an arylhydrazone fragment and a substituent at the cyclic nitrogen atom (Figure 1). The most expected types of biological action of pyrazolones were studied. First of all, we evaluated the esterase profile of the synthesized compounds, including the determination of their inhibitory activity against carboxylesterase (CES, EC 3.1.1.1) and functionally related cholinesterases—acetylcholinesterase (AChE, EC 3.1.1.7) and butyrylcholinesterase (BChE, EC 3.1.1.8), because their precursors, polyfluoroalkyl-containing 2-arylhydrazinylidene-3-oxoesters, inhibit CES [64,65,66,67]. Moreover, effective inhibitors of acetylcholinesterase were found among the non-fluorinated analogues of 4-AHPs [28]. In addition, antimicrobial, antitumor and antioxidant activities and the ability to stain cells were evaluated in vitro; analgesic and anti-inflammatory effects and acute toxicity were determined by in vivo experiments. Molecular docking was carried out to determine the mechanism of the analgesic action. The activity against the transient receptor potential cation channels A1 and V1 was observed.

## 2. Results

### 2.1. Chemistry

For the synthesis of polyfluoroalkyl-containing 4-AHPs, we utilized the two most popular approaches and found new original methods as well.

The first method for the synthesis of target 4-AHPs was based on the cyclization of ethyl 2-arylhydrazinylidene-3-polyfluoroalkyl-3-oxopropionates **1a–e** with hydrazines **2a–c** in refluxing ethanol (Figure 1). In the second method, the products were synthesized *via* the azo coupling of 3-polyfluoroalkylpyrazol-5-oles **3a–l** with (het)aryldiazonium chlorides **4a–m** according to the referring procedures [51,55]. As a result of these two approaches, a large series of 4-AHPs **5a–q, 6a–h, 7a,b, 8a,b** and **9a,b** was obtained, varying the polyfluoroalkyl moiety, the residue at the nitrogen atom N1 and an arylhydrazone fragment having substituents of a different nature (Figure 1, Table 1).

Note that only trifluoromethyl-pyrazolones **5a,b,e** among all the series of the synthesized compounds had been obtained and characterized earlier [51,59]. Heterocycles **5f,m,n** are mentioned in patent [68], but without their physicochemical characteristics. To compare the biological action, 4-tolylhydrazinylidene-5-methylpyrazol-3-ones **9a** [69] and **9b** [54] were synthesized by the known methods (Figure 1). Between these synthetic methods, the second one is obviously the most preferable because the preparation of the starting 2-arylhydrazinylidene-3-oxoesters **1** is often complicated by the formation of by-product formazans due to the acid cleavage of the dicarbonyl fragment [62,63]. It makes their isolation difficult, since a one- or two-column chromatography round followed by recrystallization is required. Pyrazolones **5–8**, obtained by the second method, do not require complex purification, since they do not undergo acid cleavage during azo coupling. Thus, we were able to increase the overall yield of compounds **5–8** up to almost quantitative level. In addition, the second method has more possibilities to synthesize various arylhydrazone derivatives based on one polyfluoroalkylpyrazolone backbone **3**.

In the reaction of ethyl 2-tolylhydrazinylidene-3-oxo-4,4,4-trifluorobutanoate **1b** with methylhydrazine sulfate **2c** in refluxing ethanol, not only the expected pyrazole **5d** was isolated, but also heterocycle **5r** having a tolyl substituent at the N2 nitrogen atom (Figure 2). Obviously, the product **5r** could be formed only upon the cyclization of two molecules of 3-oxoester **1b** accompanied by the partial decomposition of one of them. Actually, with the refluxing of ester **1b** in ethanol for 24 h in the presence of catalytic amounts of hydrochloric acid, only pyrazole **5r** was obtained. However, the incomplete conversion of the starting ester **1b** was observed under these conditions. To improve the yield of the target product **5r**, different conditions were tested by varying solvents and acid catalysts (EtOH/NaOAc; MeCN/HCl; toluene/*p*-TSA; CF_3_CH_2_OH/HCl; EtOH/TFAA or *n*-BuOH/HCl). The most efficient conditions were found to be refluxing in *n*-BuOH with HCl catalysis, which decreased the reaction time to 9 h and increased the yield to 52%. The self-condensation of pentafluoroethyl-substituted 2-tolylhydrazinylidene-3-oxoester **1c** also proceeded successfully in *n*-BuOH in the presence of HCl to give pyrazole **6i** (Figure 2).

For the reaction of ethyl 2-arylhydrazinylidene-3-polyfluoroalkyl-3-oxopropionates **1** with substituted hydrazines **2**, two regioisomers can be obtained through the initial attack of the free amino group at the polyfluoroacyl or ester fragment. However, the reactions of esters **1** with hydrazines **2** proceeded chemoselectively with the formation of one regioisomer that was confirmed by the GC-MS method, ^1^H and ^19^F NMR spectroscopy (solvent – CDCl_3_).

The 5-R^F^-regioisomeric structure of 4-AHPs **5a,b,d–f, 6a** synthesized from 2-arylhydrazinylidene-3-oxoesters **1** and hydrazines **2** was determined by the identity of products **5c,g–q, 6b–h** obtained through the second method based on pyrazoles **3a,c**, having the established structure [70,71]. Therefore, we conclude that the presence of an arylhydrazone substituent in 3-oxoester does not change the direction of cyclization with substituted hydrazines **2b,c**, which occurs *via* the condensation of the NH_2_-group of the dinucleophile at the polyfluoroacyl fragment.

We grew monocrystals for 4-AHPs **5b, 6b,g** to elucidate their structure by XRD analysis (Figure 2a–c), which confirmed their structure as 5-R^F^-regioisomers, existing in a *Z*-hydrazone amide (**HA**) tautomeric form in crystals. The stabilization of the *Z*-**HA** isomers was realized through the intramolecular hydrogen bond (IHB) formation between the proton of the NH group of the arylhydrazone substituent and the carbonyl oxygen atom of the pyrazole fragment (O2⋯H3 2.045 Å for **5b**, H4⋯O3 and H4A⋯O3A 2.043 and 2.019 Å for **6c** (two crystallographically independent molecules), (Figure 2a,b)). In pyrazolone **6g** (Figure 2c), in addition to a similar IHB (H4⋯O1 1.992 Å), the formation of intermolecular chlorine–oxygen bonds (Cl1⋯O1’ 3.177 Å) was revealed leading to molecule dimerization.

The IR spectra of all compounds **5–9** recorded for the solid state had an absorption band of the carbonyl group at ν 1652–1667 cm^–1^ indicating their existence in *Z*-**HA** form. In solutions, heterocycles **5–9** can undergo azo-hydrazone and keto-enol tautomerism, and they can exist not only as the **HA** tautomer, including *Z*- and *E*-isomers, but also as azo-enol tautomers (**AE**) (Figure 3).

The IR spectra of compounds **5b, 6c,g**, registered for a 0.1 M solution of CHCl_3_ and for the solid state had absorption bands of characteristic vibrations of carbonyl groups at ν 1660–1670 cm^–1^ that allow us to suggest the presence of the *Z*-**HA** form for pyrazoles **5–9** in solutions too. The NMR spectra of heterocycles **5–9** recorded in CDCl_3_ and DMSO-*d*_6_ contained one set of signals, which also corresponds to the *Z*-isomeric form of the **HA** tautomer, since the ^1^H NMR spectra contain downfield proton signals at δ 13–14 ppm of the NNH group that can participate in the IHB formation. Similarly, the signal of the carbon atom of the carbonyl group at δ 157–160 ppm in the ^13^C NMR spectra was specific for the **HA** tautomer. According to QC, *Z*-isomer of pyrazolone **5a** is more energetically stable than *E*-isomer. The difference in Gibbs free energy is 26 kJ/mol (Appendix A). Thus, we showed that 4-AHPs are the synthetically available compounds existing in the stable *Z*-**HA** form.

### 2.2. In Silico ADME Studies

We carried out calculations of ADME parameters for the theoretical evaluation of the physicochemical and pharmacokinetic characteristics that would allow us to predict whether the obtained compounds could be used as a remedy [72]. The pharmacological activity forecast and the drug-like level was determined using the QSAR approach in the subprogram QikProp (software Schrodinger Small-Molecule Drug Discovery Suite) [73]. The in silico ADME properties and other pharmaco-kinetical properties *viz*. the partition coefficient of *n*-octanol and water, aqua solubility, polar surface area, number of rotatable bonds, number of hydrogen bond donors/acceptors, caco-2 cell permeability, human serum albumin binding, blood/brain partition coefficient and human oral absorption of all synthesized compounds were computed by using the QikProp program. The most valuable parameters are summarized in Table 2, with special attention paid to the parameters that comply with the Lipinski and Jorgensen rules. Highlighted in red is the range of minimum and maximum recommended values (in software limits) corresponding to 95% of the known drugs.

In column 2 (#star), the total number of descriptors is indicated that go beyond the permitted values of the various physicochemical and structural parameters. The maximum allowed disagreement was limited to five. The maximum number of faults (three) was revealed for 4-[2-(2,4-dichlorophenyl)hydrazinylidene]-5-(pentafluoroethyl)-2-phenyl-2,4-dihydropyrazol-3-one **6f**. The compounds **5c, 6d,e,g, 7b, 8b** have two faults. All disagreements are connected with their structural peculiarities, which affect the excess, mainly, of two parameters—calculated electron affinity (the negative of LUMO energy and the weakly polar component of the SASA (halogens) (Appendix A).

It is evident from Table 2 that all the synthesized compounds **5–9** follow Lipinski’s rule and display favorable pharmacokinetic profiles for all descriptors. The molecular masses and calculated number of the allowed donor and acceptor hydrogen bonds of the investigated compounds with water correspond to permitted values, although the value of the donorHB parameter for most compounds is at the boundary values equal to zero. The estimation of the octanol-water partition coefficient (lipophilicity) examined by the QPlogPo/w value showed good absorption for all compounds **5–9**, excepting 5-(nonafluorobutyl)-2-phenyl-4-(2-methylphenylhydrazinylidene)-2,4-dihydropyrazol-3-one **8b**, having a border value with the admissible one. The polar surface area (PSA), which should not be more than 200 Å, is another key property linked to drug bioavailability and the values for the target molecules are in the range of 61.1–131.1 Å.

Pyrazolones **6e,f,g, 7b, 8b** do not conform to the Jorgensen rule of three according to the same parameter QplogS, indicating their low solubility in an aqueous solution. Probably, the presence of the bulky moieties in pyrazolones **6e,f,g, 7b, 8b** results in the reduced QPlogS. However, these space factors do not influence the Caco-2 cell permeability parameter (QPPcaco) for this and the other compounds **5–9**. The decrease in this parameter for pyrazolones **5m–q**, having sulfogroups, can be noted, and such a reduction is significant for pyrazolone **5o** having a benzenesulfonic acid residue. The number of possible metabolic reactions for most of the synthesized pyrazolones was no more than two, while for the derivatives with phenyl substituents **5r, 9a,b** it was three.

The blood/brain partition coefficient (QPlogBB) value, which is a measure of the ability of a drug to cross the blood–brain barrier, was in the acceptable range of –1.6 ÷ 0.5 for all molecules. Further, the human serum albumin binding co-efficient (QPlogKhsa) is also one of the key factors, and the predicted values for the compounds were in the range of –0.5 ÷ 0.9. The predicted values of the human oral absorption (HOA) of the compounds were in the range of 70–100%, which indicates the possibility of their good or high oral bioavailability. The reduction of the HOA parameter (<80%) was observed for pyrazolones **5n-p** bearing sulfonamide or benzenesulfonic acid moiety.

Thus, all the compounds excepting pyrazolones **6e,f,g, 7b, 8b** can be considered as “drug-like” and were expected to have an acceptable ADME profile.

### 2.3. Biological Evaluation

Taking into account the multiplicity of biological activity of pyrazole-containing drugs effecting various biotargets, we performed multiple in vitro and in vivo tests for fluorine-containing 4-AHPs **5–9** in comparison with the non-fluorinated analogues **9a,b**.

#### 2.3.1. Esterase Profile of 4-AHPs **5–9**

The esterase profile assessment of the synthesized pyrazolones **5–9** included a comparative evaluation of their inhibitory activity to three serine esterases: acetylcholinesterase (AChE, EC 3.1.1.7), butyrylcholinesterase (BChE, EC 3.1.1.8) and carboxylesterase (CES, EC 3.1.1.1) [74,75].

Compounds inhibiting cholinesterases are considered as potential agents for the treatment of a number of diseases associated with abnormality in cholinergic neurotransmission (Alzheimer’s disease, myasthenia gravis, etc.). The inhibitors of CESs could be used as co-drugs to improve pharmacokinetics and the efficacy and safety profiles of clinically applied drugs for which CESs are involved in their metabolism and clearance [76,77,78,79,80]. Given the critical role of human CES1 (hCES1) in metabolizing cholesteryl esters, inhibitors of hCES1 have the potential to treat hypertriglyceridemia, obesity, type 2 diabetes and atherosclerosis [80,81,82,83].

It should be noted that the inhibition of CESs can lead to undesirable drug–drug interactions with other ester-containing medicines taken by the patient [84,85]. At the same time, for CES inhibitors developed as drugs or co-drugs, the presence of anticholinesterase activity is undesirable [86]. Earlier, the nanomolar anti-CES activity of polyfluoroalkyl-containing 2-arylhydrazinylidene-3-oxoesters was found [65,67,87]. Their heterocyclization at the dicarbonyl fragment led to a weakening of the inhibitory effect by two to three times for heterocyclic azabicycles [88,89], and to activity loss for isoxazole derivatives [90]. In addition, some pyrazole and pyrazolone derivatives are known to have high anticholinesterase activity [28,91]. Therefore, it is of interest to study new synthesized pyrazolones as inhibitors of CES, AChE and BChE.

The study of the esterase profile of 4-AHPs **5–9** (Table 3) showed that these pyrazole derivatives were not active or exhibited weak inhibitory activity against AChE and BChE, while some of them demonstrated inhibitory activity towards CES. The performed research allowed us to identify a group of fluorine-containing 4-AHPs **5a,d-h,j,l,r, 6b-d,g, 7b, 8a** as selective CES inhibitors, and five of them (**5a,e,f,l** and **6b**) showed potent activity in the submicromolar range of IC_50_ from 0.248 μM to 1.01 μM. In particular, CF_3_-pyrazolones **5a** and **5e** bearing a phenylhydrazone substituent have the maximum activity with IC_50_ of 0.248 and 0.481 μM. The introduction of an ethoxycarbonyl group into position 4 of the aryl substituent (compound **5l**) led to an increase in the anti-CES activity. In the C_2_F_5_ series of pyrazolones **6**, the most active was derivative **6b,** having tolylhydrazone moiety.

The rest of the compounds, **5b,c,i**,**k**,**m–r**, **6a,e**,**f,h**, **7a**, **8b, 9a,b,** did not reveal significant anti-esterase activity, which indicates the absence of the possible undesirable side effects associated with the inhibition of CESs and cholinesterases in their other biomedical applications.


*Kinetic Studies of CES Inhibition*


The inhibition kinetics of CES were studied for active compounds **5a,d,f**. The obtained values of the inhibition constants (*K*_i—_competitive component and α*K*_i_—non-competitive component) are shown in Table 4 and demonstrate a mixed type of inhibition.

Thus, a series of compounds with potent anti-CES activity were identified among 4-AHPs **5–9**. The lack of anticholinesterase activity of these compounds allows us to consider them as promising candidates for modulating the rate of hydrolytic metabolism and the rational use of esterified drugs, as well as their use for studying the physiological role of CES. At the same time, compounds **5a,d–h,j,l,r, 6b–d,g, 7b, 8a,** having anti-CES inhibitory activity, should be suggested for other medical purposes with caution due to possible undesirable drug–drug interactions.

#### 2.3.2. Antioxidant Activity

The high antioxidant activity of arylhydrazinylidenepyrazolones containing coumarin moiety in the DPPH (2,2-diphenyl-1-picrylhydrazyl) test [12] and N-thiazolyl derivatives in the ABTS (2,2′-azinobis-(3-ethylbenzothiazoline-6-sulfonic acid)) test [32] has been described. However, the antioxidant properties of the 4-arylhydrazinylidene-5-methylpyrazol-3-one core, especially of its polyfluoroalkyl analogues, have not yet been evaluated, although the structural similarity of compounds **5–9** with the known antioxidant Edaravone suggests radical-scavenging property presence.

We evaluated the primary antioxidant activity of synthesized 4-AHPs **5–9** in comparison with their non-fluorinated analogues **9a,b** using three standard tests: ABTS, FRAP (ferric reducing antioxidant power) and ORAC (oxygen radical absorbance capacity) assays. The ABTS assay evaluates the binding of a model ABTS radical cation (ABTS^•+^), which is realized by the mechanism of single electron transfer (SET) and/or hydrogen atom transfer (HAT). The FRAP assay measures the ability of compounds to reduce the ferric 2,4,6-tripyridyl-s-triazine complex [Fe(TPTZ)_2_]^3+^ to [Fe(TPTZ)_2_]^2+^, which occurs exclusively by the SET mechanism. The ORAC-FL assay measures the oxidative degradation of the fluorescent molecule (FL – fluorescein) after being mixed with a free peroxyl radical generator such as 2,2′-azobis-(2-methylpropionamidine) dihydrochloride (AAPH) in the presence of antioxidant (tested compound). The ORAC-FL test describes the antioxidants’ ability to yield the hydrogen atom and, consequently, it is a HAT-based assay [92]. Trolox was used in all tests as a reference antioxidant: the antioxidant activity of the test compounds was referred to the activity of Trolox. The well-known antioxidant Edaravone was also used for comparison. The results are presented in Table 5.

The ability of NH-unsubstituted 4-AHPs **5a–c, 6a, 7a, 8a** to bind the ABTS^•+^ radical was shown to be quite high and equal to about half the activity of Trolox. In contrast, N-aryl-substituted analogues **5d–r** showed weak antiradical activity or none. The noticeable radical-scavenging activity of the N-Ph-substituted compound **5g** could be associated with the presence of an electron-donor MeO-substituent in the *para*-position of the arylhydrazone fragment.

The non-fluorinated pyrazolones **9a,b,** independently of the presence of a substituent at the nitrogen atom, demonstrated high radical-scavenging activity (TEAC 0.86 and 0.8, respectively), close to Trolox and Edaravone. It should be noted that compound **9b** had the closest structure to Edaravone.

All the compounds **5–9** were less active in the FRAP test. The NH-pyrazolones **5a,b, 6a,** which have a “short” polyfluoroalkyl substituent, showed a slight iron-reducing activity. Only the non-fluorinated NH-pyrazolone **9a** demonstrated activity in the FRAP test comparable with Trolox and Edaravone, while the activity of N-phenyl-5-methylpyrazolone **9b** was significantly lower.

The results of the ORAC-FL test indicate that NH-unsubstituted pyrazolones **5a–c, 6a, 7a, 8a,** compound **5d** containing the electron donating methyl group at the N atom, and the only derivative of the series of N-aryl-substituted pyrazolones (heterocycle **5g**) bearing the methoxy group in the *para*-position of the phenylhydrazone fragment have antioxidant activity. The other N-aryl-substituted analogues either did not show antioxidant activity (substances **5h,r, 6b, 7b, 8b**) or showed a pronounced decrease in the fluorescence of fluorescein to values below background ones. This property was revealed for compounds with different substituents in the arylhydrazone fragment (**5e,f,i,k,l**), as well as for all the pyrazolones **5m–q** and **6h** containing a sulfonic group. Perhaps an additional interaction of the substances with the components of the test system occurs during the test. This makes it impossible to interpret the results in terms of the presence or absence of an antioxidant effect. The non-fluorinated pyrazolones **9a,b** showed significant radical-scavenging activity in the ORAC-FL test, and the NH derivative **9a** was the most active among the tested compounds.

Comparing the obtained experimental data, we can note the agreement between the values of compound antioxidant activity in the ORAC and ABTS tests (see Table 5) in contrast to their activity in the FRAP test. Given that the FRAP test works only by the SET mechanism, and ORAC—only by HAT, it can be assumed that the radical-scavenging effect of pyrazolones **5–9** in the ABTS test is realized by the HAT mechanism. In other words, the studied pyrazolones are more prone to radical-scavenging action by the HAT mechanism.

#### 2.3.3. Antimicrobial Activity

A number of publications have been devoted to the design of antibacterial and antifungal agents based on the 4-AHP core in recent years [10,11,12,13,14]. At the same time, we found only two papers describing the antimicrobial properties of 5-trifluoromethyl-4-AHP derivatives having a β-D-glycoside residue [51] or the second N-pyrazole moiety [52]. However, the influence of the 4-AHP skeleton itself on this activity is still unclear.

According to the WHO statement, a new antimicrobial agents’ development is a high-importance task. Therefore, we tested the antimicrobial activity of the molecules against strains of three pathogenic dermatophytes (*Trichophyton rubrum*, *Epidermophyton floccosum*, *Microsporum canis*), one strain of yeast-like fungi, *Candida albicans*, and one strain of the clinically significant obligate and opportunistic bacteria *Neisseria gonorrhoeae*. The minimum inhibitory concentration (MIC) data, which characterized the power of inhibition of fungi growth, are presented in Appendix A. Fluconazole was used as positive control. Among all the compounds, only pyrazolone **5c** combining the bromine atom in the aryl substituent and the NH fragment showed significant antifungal activity at MIC 12.5 μg/mL against the dermatophytes *T. rubrum* and *E. floccosum* and at 25 μg/mL against *M. canis*.

In contrast, the antibacterial activity was more common for the studied heterocycles **5–9** when the *N. gonorrhoeae* bacteria were selected for measuring (Table 6). Thus, pyrazolones **5g**, **6b** showed a high anti-gonorrhea effect at MIC 0.9 μg/mL, which was significantly higher than the activity of the reference drug Spectinomycin. Compounds **5d**, **6d**,**g** (MIC 15.6 μg/mL) and **5j,k,l**, **7b** (MIC 31.2 μg/mL) demonstrated moderate activity. The growth of *N. gonorrhoeae* was weakly inhibited by pyrazolones **5b,f,o,p, 6a,e** at MIC 62.5 µg/mL and compounds **5c,h,m** at MIC 125 µg/mL. At the same time, a clear influence of structural factors on the antimicrobial effect was not revealed, although it was possible to notice an increase in the activity of the Ph-N substituted derivatives **5g, 6b**.

#### 2.3.4. Cytotoxicity

Recent investigations have shown that 4-AHPs are also promising for the development of antiproliferative agents [18] with different mechanisms of action [15,16,17,19]. Anticancer activity has been found for some trifluoromethyl-4-AHP nucleosides [48,49]. However, the effect of trifluoromethyl and especially the polyfluoroalkyl substituent in 4-AHPs on tumor cells has not yet been studied.

This gap is filled in this work. We studied the cytotoxic activity of compounds **5b,f,m,p,q, 6a,b, 7a, 8b, 9a,b** on a transplantable culture of human cervical carcinoma HeLa cells and a culture of human dermal fibroblasts of five to seven passages, with Doxorubicin and Camptothecin used as reference drugs. The IC_50_ values and the selectivity index (SI) were determined for the compounds having cytostatic activity (Table 7).

The NH-pyrazolones **5b, 6a** and **9a** (IC_50_ 1.47–3.16 μM) exhibited the highest cytotoxic activity against HeLa tumor cells, and the fluorine-containing derivatives **5b** and **6a** were more active than the non-fluorinated analogue **9a**. The most active cytostatic was C_2_F_5_-pyrazolone, having an IC_50_ at the level of Camptothecin. For the aryl-N-pyrazolones, the CF_3_ derivative **5f** showed activity comparable to Doxorubicin, while compounds **7b, 9b** were by an order less active (Table 7).

Interestingly, the introduction of sulfonyl groups into pyrazolones **5m,p,q** led to a complete loss of antitumor activity on the studied cells, although pyrazolones containing sulfonyl substituents in the arylhydrazone fragment are considered promising protein tyrosine phosphatase SHP2 inhibitors [15]. The elongation of the polyfluoroalkyl substituent also had a negative effect on the antitumor activity, since pyrazolone **8b** with a nonafluorobutyl substituent turned out to be inactive compared to analogues **5b, 6b, 7b** bearing “shorter” fluorinated residue.

Morphological changes in cells under the influence of compounds **5b,f, 6a, 7a, 9a,b** were detected that indicate the development of destructive processes (Figure 3). In all cases where the destruction of the cytoplasmic membrane occurs, pyknosis of the nuclei or their swelling was observed. This indicates the development of necrosis or apoptosis. Phenyl-N-CF_3_-pyrazolone **5f** had the strongest effect, leading to complete cell destruction.

For normal human fibroblasts, N-unsubstituted CF_3_-pyrazolone **5b** showed the highest cytotoxicity; its Ph-substituted analogue **5f** demonstrated moderate cytotoxicity with the same activity as compounds **6a,b, 7b** and **9a,b**. All the studied pyrazolones had a cytotoxicity less than the reference drugs. The pyrazolones **5m,p,q** and **8b** without antitumor activity did not have a cytotoxic effect on normal human fibroblasts either (Table 7).

The study showed that the 4-AHP backbone is a very attractive scaffold to develop anticancer agents. Among the tested compounds, pyrazolones **6a** and **9a** demonstrated the best selective effect (SI 8.83 and 7.68, respectively) for fibroblast/HeLa cells that exceeded the Doxorubicin and Camptothecin effect (SI 4.98 and 0.64).

#### 2.3.5. Live Cell Visualization

Tartrazine, having 4-(4-sulfonatophenylazo)-5-pyrazolone moiety, is widely used as a synthetic lemon-yellow azo dye for food coloring [93]. Many reports about the pyrazolone-based dyes for different purposes have been published to date [94,95,96]. However, there are no reports describing 4-AHPs’ use for the biovisualization of living cells. In this work, we studied the potential of compounds **5f, 6a, 9a** for the fluorescent visualization of the Vero cell structure (African green monkey kidney epithelium). Cells stained with the compound at a concentration of 1 × 10^–4^ M in DMEM were examined using an LSM-710 laser scanning confocal microscope.

All the studied compounds were found to have similar fluorescent properties and distribution among the cellular compartments (Figure 4 and Appendix A). Fluorescence was observed upon excitation by lasers with the wavelengths 405, 488, 514 and 561 nm. However, the fluorescence intensity of molecules in cells was rather low for obtaining high-resolution images and for the unambiguous determination of substance localization. We found the heterogeneous distribution of substances in cells. First of all, bright fluorescent rounded granules stand out, which can be identified as endosomes of a different nature or lysosomes. In addition, a uniform less intensive fluorescence was detected overall in cells, apparently produced by the nonspecific accumulation of substances in the cytoplasm. The brightest images for compound **6a** are presented (Figure 4).

It is interesting that the fluorescence intensity of this substance was sufficient to obtain a relatively high-quality image by a 561 nm laser excitation followed by the unequal emission in two spectral ranges. The cell structures near the nucleus (likely the Golgi apparatus or part of the endoplasmic reticulum) had an emission in the green-yellow range (561–580 nm), while in the red range (600–650 nm) the emission of granular round formations was monitored (Figure 5).

As seen from the images, the 4-AHPs **5f, 6a, 9a** can be accumulated in Vero cells, and endosomes or lysosomes accumulated more molecules. No acute phototoxicity of the substances, which could occur in photochemical reactions of a radical or other nature, was observed during the experiments.

#### 2.3.6. Analgesic and Anti-Inflammatory Activity


*Analgesic activity*


Some 4-AHP derivatives have shown great potential as analgesic and anti-inflammatory agents [11,13,30,31], especially 4-aminosulfonyl-containing [30] and 5-methyl-4-[(*Z*)-(4-nitrophenyl)diazenyl]-2,4-dihydro-3H-pyrazol-3-one [31] derivatives, which were found to be the most potent COX/5-LOX inhibitors. Therefore, the analgesic and anti-inflammatory activity of the synthesized polyfluoroalkyl-containing 4-AHPs could be expected.

The tested 4-AHPs were measured on analgesia but initially we evaluated the acute toxicity of compounds **5–9** in a dose of 300 mg/kg on the minimum recommended number of mice (three animals [97]) at intraperitoneal (*ip*) administration. For all the molecules, a 100% survival of the animals was observed (Table 8). At the same time, it was found that all the tested substances were less toxic than diclofenac, which is widely used to relieve various pain syndromes. Note that the LD_50_ of diclofenac has a very low value of 74 mg/kg [98], which was confirmed by the low viability (66%) of mice at a dose of 100 mg/kg compared to metamizole and 4-AHPs.

Then, we studied the analgesic activity of 4-AHPs **5–9** in the hot plate test on SD rats at a dose of 15 mg/kg, using metamizole and diclofenac as reference drugs (Table 8). The obtained results confirmed an idea for developing effective analgesics on the base of fluorinated pyrazolone derivatives, since a lot of studied molecules showed some antinociceptive effect over one to two hours. The pyrazolone **5d**, containing a methyl substituent at the ring nitrogen atom, and pyrazolones **5j, 6e,f**, having two chlorine atoms in the aryl substituent of the hydrazone fragment, were inactive in the test. At the same time, their analogues containing one chlorine atom in position 2 of the aromatic ring (**5i** and **6c**) or in position 3 (**6d**) showed pronounced analgesic activity.

The “structure-activity” relationship analysis showed that the nature of the substituent at the nitrogen atom of the cycle significantly affects the antinociceptive effect. Thus, HN-pyrazolones were less active compared to N-phenyl-substituted analogues (**5a** vs. **5e**, **5c** vs. **5k**, **6a** vs. **6b**, **8a** vs. **8b** and **9a** vs. **9b**). Only compounds **5b** and **5f,** showing approximately the same level of activity, stand out from this series. The introduction of a methyl substituent at the N-position of the pyrazole ring leads to a complete loss of the analgesic effect of compound **5d**.

The CF_3_-pyrazolone **5e** containing two phenyl substituents showed a pronounced analgesic effect exceeding the activity of the reference drugs. The introduction of a methyl or methoxy group, as well as fluorine, chlorine, or bromine atoms into the *para*-position of the arylhydrazone fragment of 5-CF_3_-2-phenylpyrazol-3-ones **5f–i, k,** led to a decrease in their antinociceptive effect. The introduction of a second chlorine atom into an arylhydrazone substituent (compound **5j**) abolished the analgesic activity. The 4-ethoxycarbonyl-substituted pyrazolone **5l** demonstrated a good analgesic effect, but weaker than expressed by the phenyl analogue **5e**. The introduction of the 4-methylsulfonyl group was the most effective, since compound **5m** showed significant analgesic activity both at 1 and 2 h of measurement. The replacement of this group with a sulfonic acid residue led to a partial loss of activity of pyrazolone **5o** in the 2nd h after administration, while its replacement with a sulfonamide fragment in compound **5n** resulted in a significant decrease in activity.

The CF_3_-pyrazolone **5r** having two tolyl substituents showed activity at the level of the N-phenyl analogue **5f**. Similarly to substitution strategy in hydrazone fragment, the introduction of a 4-methylsulfonyl group into the N-phenyl fragment was effective, and compound **5q** had a strong analgesic effect at both measurement points in contrast to the sulfonamide analogue **5p**.

Among C_2_F_5_-pyrazolone **6**, 4-tolylhydrazinylidene-2-phenylpyrazolone **6b** showed extraordinarily high activity at 1 h after injection, exceeding the effect of the CF_3_ analogue **5f**. Compound **6h,** combining C_2_F_5_ and 4-MeSO_2_ substituents, showed a pronounced analgesic action, but less than the CF_3_ analogue **5m**. The effect of other substituents in C_2_F_5_-pyrazolones **6c–g** on their antinociceptive properties had a similar trend as for the CF_3_-derivatives **5i,j,l**.

In the series of C_3_F_7_- and C_4_F_9_-containing pyrazolones, N-Ph-derivatives **7b** and **8b** were more active than HN-analogues **7a** and **8a** at both measurement points. At the same time, pyrazolone **7b** showed a superior analgesic effect at 1 h after administration, being inferior to the activity of the C_2_F_5_ analogue **6b** only. Pyrazolone **8b** had a strong effect at 2 h, which was weaker than the activity at this point of CF_3_-compounds **5m** and **5e** only.

The length of the polyfluoroalkyl substituent does not critically affect the activity of the N-unsubstituted 4-tolylhydrazinylidene-derivatives **5b, 6a, 7a** and **8a**, although the presence of C_2_F_5_ (compound **6a**) may be considered as to some extent favorable. For the set of N-phenyl-4-tolylhydrazinylidene-derivatives, the elongation of the polyfluoroalkyl residue significantly enhances the antinociceptive effect: compounds **6b**, **7b** and **8b** are substantially more active than the CF_3_-analogue **5f**.

The non-fluorinated pyrazolones **9a** and **9b** had rather high activity, but only for 1 or 2 h of measurement, respectively.

Thus, compounds with various combinations of substituents in the hydrazone fragment and at the N atom of the pyrazole ring were found to be very promising for further optimization. The leading compounds showing significant analgesic activity at both time points were CF_3_-pyrazolones **5e,m,q** and C_4_F_9_-pyrazolone **8b**. In addition, phenyl-N-substituted C_2_F_5_- and C_3_F_7_-pyrazolones **6b**, **7b** demonstrated the highest activity at 1 h of the test, and the Me-analogue **9b** was active at the level of pyrazolone **5m** at 2 h after administration.


*Anti-inflammatory action*


Unlike classical non-steroidal anti-inflammatory drugs (NSAIDs) with COX-1 and/or COX-2 as the main target, the known pyrazolone analgesic metamizole (dipyrone) can only slightly inhibit COX and therefore does not show an anti-inflammatory effect [99,100]. We tested whether synthesized compounds have an additional extremely useful property—anti-inflammatory activity. The measurements were performed at 1, 3, 5 and 24 h after carrageenan injection (Table 9). The classic NSAID, diclofenac, confirmed its pronounced effect in this model at a dose of 10 mg/kg. However, all the tested compounds were inactive at a dose of 15 mg/kg at all the time points, except **5f** showing some activity at the 5^th^ h. Besides, the methylsulfonyl derivative **5m,** which was tested at a higher dose of 25 mg/kg, showed an anti-inflammatory effect.


*Determination of a mechanism of analgesic action*


The absence of an anti-inflammatory response together with a pronounced analgesic activity resembles the activity spectrum of metamizole, which is similar in structure to the 4-AHPs **5–9**. The mechanism of the analgesic action of metamizole has not been thoroughly studied; however, transient receptor potential ankyrin 1 (TRPA1) has been named as one of the proposed targets of its superior analgesic action [99].

TRPA1 is known to be responsible for inflammatory and neuropathic pain appearance; therefore, it can be considered as a potential biological target for the analgesic action of the studied pyrazolones [101,102,103]. Therefore, we studied the effect of pyrazolone **6b,** showing the maximum analgesic effect in the hot plate test at 1 h on the activation of the TRPA1 ion channel by allyl isothiocyanate (AITC). The cell line CHO (Chinese hamster ovary cell) stably expressing the TRPA1 channel was used for measurement. Activation by a specific agonist was measured by fluorescence spectroscopy using Fluo-4 dye by changing the level of intracellular calcium. However, compound **6b** had no significant effect on AITC-induced TRPA1 channel activation.

Noxius temperature detection is carried out at the molecular level using another important transient receptor potential vanilloid 1 (TRPV1) receptor. This multimodal receptor can be activated by acid, temperature or chemical agents (selective agonist capsaicin), and it is recognized as one of the main receptors for the nociception system [104]. The molecules reducing temperature sensitivity in the hot plate test may be antagonists of this receptor, and we tested the effect of pyrazolone **6b** on TRPV1 ion channel activation in CHO-TRPV1 cells.

The multimodality of TRPV1 activation was tested by adding an acidic solution or 160 μM solution of 2-aminoethoxydiphenyl borate (2APB) agonist (Figure 6A). A specific inhibitor of the TRPV1 ion channel—capsazepine in 50 μM concentration—effectively inhibited the increase in intracellular Ca^2+^ in both cases. The pre-incubation of the cells with compound **6b** significantly reduced the recorded cell response. In three independent experiments, the activation of the ion channel by acidification (Figure 6B) at 50 μM concentration of **6b** decreased the response by up to 47, 62 and 43% compared to control (2% (*v/v*) DMSO, pre-incubation). Similarly, TRPV1 activation by acidification together with 80 μM 2APB (Figure 6C) were reduced by **6b** by up to 54, 36 and 44% compared to control. Therefore, **6b** is an inhibitor of the TRPV1 ion channel and other 4-AHPs may share some inhibitory activity to TRP ion channels as well; this should be tested in further experiments.

### 2.4. Molecular Docking

To determine the action mechanism of **6b** as a TRPV1 channel antagonist, we performed its molecular docking in the vanilloid binding site.

The molecular docking procedure was carried out using induced fit docking protocol (flexible ligand and protein for the stable *Z*-isomer of pyrazolone **6b**). The possibility of the deprotonation of **6b** was taken into account because the value of the acidity constant (pKa) is equal to 7.31; therefore, at physiological pH the molecule presents in deprotonated and neutral forms. Both molecular forms of **6b** were subjected to molecular docking, and according to the molecular docking results both forms can interact with the vanilloid binding site of TRPV1.

In this case, its *para*-tolyl substituent is located in the same binding cavity as the aromatic fragment of capsaicin (Figure 7A,B) with the formation of π-π staking interaction with Tyr^513^. The pentafluoroethyl group is located in the hydrophobic part of the binding site, with possible interaction with the side chains of amino acids such as Ile^539^, Leu^579^, Ala^667^, Leu^671^. In this case, this fragment of the molecule is located at the boundary of the binding site, so it can interact with the inner lipophilic part of the cell membrane. The phenyl fragment is also characterized by the position in the hydrophobic part of the binding pocket formed by Phe^545^, Ala^548^, Phe^593^, Leu^664^, with the formation of π-π stacking interactions with the amino acids Phe^545^ and Phe^593^ (Figure 7C).

The deprotonated form of pyrazolone **6b** is positioned somewhat differently, and a hydrogen bridge is fixed between the carbonyl oxygen of the ligand and Tyr^513^. The energy parameters such as docking score (first number in Figure 7C) and clustering energy (second number in Figure 7C) are markedly higher for the deprotonated form (Figure 7D). Probably, neutral forms of pyrazolone **6b** make a greater contribution to the binding of the ligand at the binding site.

Based on theoretical studies, it can be concluded that the determining inhibitory activity is the binding of pyrazolone **6b** both to the intraprotein part of the TRPV1 channel pocket and to its lipophilic surface directly interacting with the cell membrane. Here, we can propose that the binding of the leading compound **6b** may influence the activation of the receptor.

## 3. Material and Methods

### 3.1. Chemistry

The solvents (acetone, acetonitrile, chloroform, hexane, ethanol, *n*-butanol, methylene chloride), hydrochloric acid, hydrazine monohydrate and phenylhydrazine were obtained from AO “VEKTON” (St. Petersburg, Russia). Methylhydrazine sulphate, sodium nitrite and sodium acetate were obtained from AO “REACHEM” (Moscow, Russia). Methyl hydrazine was purchased from Merck KGaA (Darmstadt, Germany). Arylamines for synthesis of aryldiazonium chlorides **4a–m** were obtained from Alfa Aesar *via* Thermo Fisher Scientific (Kandel, Germany). The deuterosolvents CDCl_3_ and DMSO-*d*_6_ were acquired from «SOLVEX» Limited Liability Company (Skolkovo Innovation Center, Moscow, Russia). Melting points were measured in open capillaries on a Stuart SMP30 (Bibby Scientific Limited, Staffordshire, UK) melting point apparatus and were uncorrected. The IR spectra were recorded on a PerkinElmer Spectrum Two FT-IR spectrometer (Perkin-Elmer, Waltham, MA, USA) using the frustrated total internal reflection accessory with a diamond crystal. The ^1^H and ^19^F NMR spectra were registered on a Bruker DRX-400 spectrometer (400 or 376 MHz, respectively) or a Bruker Avance^III^ 500 spectrometer (500 or 470 MHz, respectively) (Bruker, Karlsruhe, Germany). The ^13^C NMR spectra were recorded on a Bruker Avance^III^ 500 spectrometer (125 MHz). The internal standard was SiMe_4_ (for ^1^H and ^13^C NMR spectra) and C_6_F_6_ (for ^19^F NMR spectra). The microanalyses (C, H, N) were carried out on a PerkinElmer PE 2400 series II (PerkinElmer, Waltham, MA, USA) elemental analyzer. The elemental analysis (C, H, N, S) for compounds **5m–q, 6h** was obtained on a CHNS Euro EA 3000 (Eurovector Instruments, Pavia, Italy). The column chromatography was performed on silica gel 60 (0.062–0.2 mm) (Macherey-Nagel GmbH & Co KG, Duren, Germany).

The initial 2-arylhydrazinylidene-3-oxoesters **1a–e** [67,105] and pyrazololes **3a–l** [106] were synthesized by referring to previously published methods.

#### 3.1.1. Synthesis of Compounds **5a–s, 6a–i, 7a,b 8a,b, 9a,b** (General Procedure)

**Method A**. The corresponding hydrazine **2a–c** (3 mmol) was added to a solution of 2-arylhydrazinylidene-3-polyfluoroalkyl-3-oxoester **1a–e** (3 mmol) in ethanol (15 mL). The resulting mixture was refluxed for 8 h. The precipitate was filtered off. The product was purified by recrystallization from ethanol.

**Method B**. An aqueous solution of sodium nitrite (NaNO_2_ 0.35 g, H_2_O 2 mL) was added dropwise at 0 °C to a solution of aromatic amine (3 mmol) in diluted hydrochloric acid (1.8 mL HCl, 5.3 mL H_2_O). At the same time, a solution of sodium acetate (2.3 g) in water (3 mL) and polyfluoroalkyl-containing pyrazole (3 mmol) **3a–l** in acetone (20 mL) were mixed in a glass. A solution of the diazonium salt **4a–m** was slowly added to the resulting suspension of pyrazole at 10 °C. The precipitate was filtered off. The product was purified by recrystallization from ethanol.

**Method C**. To a solution of 2-arylhydrazinylidene-3-polyfluoroalkyl-3-oxo ester **1b, c** (3 mmol) in *n*-butanol (15 mL) was added 2 mL of concentrated HCl. The resulting mixture was refluxed for 12 h. Then, the mixture was cooled to room temperature. The precipitate was filtered off. The product was purified by recrystallization from ethanol.

*(Z)-4-(2-Phenylhydrazinylidene)-5-(trifluoromethyl)-2,4-dihydro-3H-pyrazol-3-one* (**5a**). Yield 0.45 g (58% *method A*); orange powder (EtOH); mp 185–186 °C (lit. [51] mp 182–184 °C).

*(4Z)-4-[2-(4-Methylphenyl)hydrazinylidene]-5-(trifluoromethyl)-2,4-dihydro-3H-pyrazol-3-one* (**5b**). Yield 0.48 g (59% *method A*); orange powder; mp 184–186 °C (lit. [51] mp 187 °C).

*(4Z)-4-[2-(4-Bromophenyl)hydrazinylidene]-5-(trifluoromethyl)-2,4-dihydro-3H-pyrazol-3-one* (**5c**). Yield 0.79 g (78% *method B*); orange powder; mp 235–236 °C; IR ν 3215, 3174 (NH, CH), 1663 (C=O), 1549, 1526, 1479, 1442 (NH, C=C, C=N), 1249–1132 (CF) cm^–1^; ^1^H NMR (CDCl_3_, 400 MHz) δ 7.36–7.39, 7.57–7.59 (4H, both d, *J* = 8.9 Hz, C_6_H_4_), 9.14 (1H, s, NH), 13.71 (1H, br. s, NNH); ^13^C NMR (CDCl_3_, 125 MHz) δ 118.16, 119.24 (q, *J* = 272.9 Hz, CF_3_), 120.67, 122.81, 133.02, 139.14 (q, *J* = 39.2 Hz, C–CF_3_), 139.42, 159.48; ^19^F NMR (CDCl_3_, 376 MHz) δ 97.09 (s, CF_3_); Anal. calcd. for C_10_H_6_BrF_3_N_4_O. C, 35.84; H, 1.80; N, 16.72. Found: C, 35.77; H, 1.69; N, 16.73.

*(4Z)-2-Methyl-4-[2-(4-methylphenyl)hydrazinylidene]-5-(trifluoromethyl)-2,4-dihydro-3H-pyrazol-3-one* (**5d**). Yield 0.38 g (45% *method A*); yellow powder; mp 146–147 °C; IR ν 3310, 3125, 3050, 2949 (NH, CH), 1658 (C=O), 1556, 1521, 1491 (NH, C=C, C=N), 1287–1174 (CF) cm^–1^; ^1^H NMR (CDCl_3_, 500 MHz) δ 2.38 (3H, s, CH_3_), 3.53 (3H, s, NCH_3_), 7.24 (2H, d, *J* = 8.4 Hz, C_6_H_4_), 7.39 (2H, d, *J* = 8.4 Hz, C_6_H_4_), 13.87 (1H, br. s, NNH); ^19^F NMR (CDCl_3_, 470 MHz) δ 97.70 (s, CF_3_); Anal. calcd. for C_12_H_11_F_3_N_4_O. C, 50.71; H, 3.90; N, 19.71. Found: C, 50.91; H, 3.95; N, 19.79.

*(4Z)-2-Phenyl-4-(2-phenylhydrazinylidene)-5-(trifluoromethyl)-2,4-dihydro-3H-pyrazol-3-one* (**5e**). Yield 0.61 g (61% *method A*); yield 0.81 g (81% *method B*); orange crystals, mp 152–153 °C (lit. [59] 148–150 °C).

*(4Z)-4-[2-(4-Methylphenyl)hydrazinylidene]-2-phenyl-5-(trifluoromethyl)-2,4-dihydro-3H-pyrazol-3-one* (**5f**). Yield 0.63 g (62% *method A*); yield 0.81 g (78% *method B*); orange powder; mp 155–156 °C; ^1^H NMR (CDCl_3_, 400 MHz) δ 2.40 (3H, s, CH_3_), 7.26–7.28, 7.41–7.49, 7.93–7.95 (9H, all m, Ph and C_6_H_4_), 14.06 (1H, br. s, NNH); ^13^C NMR (CDCl_3_, 125 MHz) δ 21.15, 116.84, 119.36, 119.62 (q, *J* = 271.1 Hz, CF_3_), 122.75, 123.26, 126.34, 129.08, 130.47, 137.42, 137.99, 138.21 (q, *J* = 32.2 Hz, C–CF_3_), 157.18; ^19^F NMR (CDCl_3_, 376 MHz) δ 97.60 (s, CF_3_); Anal. calcd. for C_17_H_13_F_3_N_4_O. C, 58.96; H, 3.78; N, 16.18. Found: C, 58.81; H, 3.52; N, 16.48.

(*4Z)-4-(2-(4-Methoxyphenyl)hydrazinylidene)-2-phenyl-5-(trifluoromethyl)-2,4-dihydro-3H-pyrazol-3-one* (**5g**). Yield 0.80 g (74% *method B*); orange powder; mp 145–146 °C; IR ν 3111, 3081, 2994, 2942 (NH, CH), 1658 (C=O), 1553, 1536, 1483 (NH, C=C, C=N), 1251–1116 (CF) cm^–1^. ^1^H NMR (CDCl_3_, 500 MHz) δ 3.86 (3H, s, OCH_3_), 6.98–7.00, 7.27–7.30, 7.45–7.49, 7.94–7.95 (9H, all m, Ph and C_6_H_4_), 14.18 (1H, br. s, NNH). ^13^C NMR (CDCl_3_, 125 MHz) δ 55.62, 115.17, 118.39, 119.31, 119.67 (q, *J* = 271.1 Hz, CF_3_), 122.23, 126.26, 129.05, 133.83, 137.45, 137.99 (q, *J* = 39.1 Hz, C–CF_3_), 157.22, 159.34; ^19^F NMR (CDCl_3_, 470 MHz) δ 97.67 (s, CF_3_). Anal. calcd. for C_17_H_13_F_3_N_4_O_2_. C, 56.36; H, 3.62; N, 15.46. Found: C, 56.46; H, 3.65; N, 15.49.

*(4Z)-4-[2-(4-Fluorophenyl)hydrazinylidene]-2-phenyl-5-(trifluoromethyl)-2,4-dihydro-3H-pyrazol-3-one* (**5h**). Yield 0.88 g (84% *method B*); orange powder; mp 135–137 °C; IR ν 3302, 3088, 3038 (NH, CH), 1661 (C=O), 1557, 1486 (NH, C=C, C=N), 1227–1137 (CF) cm^–1^; ^1^H NMR (CDCl_3_, 500 MHz) δ 7.16–7.19, 7.30–7.31, 7.45–7.52, 7.92–7.94 (9H, all m, Ph and C_6_H_4_), 14.03 (1H, br. s, NNH); ^13^C NMR (CDCl_3_, 125 MHz) δ 116.95 (d, *J* = 23.4 Hz, C(3’)_Ar_–F), 118.43 (d, *J* = 8.3 Hz, C(2’)_Ar_), 119.30 (q, *J* = 271.1 Hz, CF_3_), 119.31, 123.27, 126.47, 129.11, 136.63 (d, *J* = 2.9 Hz, C(1’)_Ar_), 137.25, 138.17 (q, *J* = 39.2 Hz, C–CF_3_), 157.06, 161.59 (d, *J* = 248.9 Hz, C(4’)_Ar_–F); ^19^F NMR (CDCl_3_, 470 MHz) δ 48.68–48.74 (1F, m, F), 97.54 (3F, s, CF_3_); Anal. calcd. for C_16_H_10_F_4_N_4_O. C, 54.86; H, 2.88; N, 16.00. Found: C, 54.56; H, 2.85; N, 16.86.

*(4Z)-4-[2-(2-Chlorophenyl)hydrazinylidene]-2-phenyl-5-(trifluoromethyl)-2,4-dihydro-3H-pyrazol-3-one* (**5i**). Yield 0.94 g (86% *method B*); orange powder; mp 179–180 °C; IR ν 3319, 3081 (NH), 1664 (C=O), 1551, 1467 (NH def., C=C, C=N), 1290–1126 (CF) cm^–1^; ^1^H NMR (CDCl_3_, 400 MHz) δ 7.23–7.25, 7.26–7.32, 7.39–7.41, 7.43–7.49, 7.90–7.96 (9H, all m, Ph and C_6_H_4_), 14.15 (1H, s, NNH); ^13^C NMR (CDCl_3_, 125 MHz) δ 116.86, 119.36, 119.42 (q, *J* = 271.2 Hz, CF_3_), 122.89, 125.00, 126.52, 127.54, 128.43, 129.13, 129.98, 137.12, 137.14, 138.13 (q, *J* = 39.3 Hz, C–CF_3_), 156.77; ^19^F NMR (CDCl_3_, 376 MHz) δ 97.47 (s, CF_3_); Anal. calcd. for C_16_H_10_ClF_3_N_4_O. C, 52.40; H, 2.75; N, 15.28. Found: C, 52.65; H, 2.53; N, 15.72.

*(4Z)-4-[2-(2,6-Dichlorophenyl)hydrazinylidene]-2-phenyl-5-(trifluoromethyl)-2,4-dihydro-3H-pyrazol-3-one* (**5j**). Yield 0.95 g (79% *method B*); orange powder; mp 133–135 °C; IR ν 3320, 3075 (NH), 1666 (C=O), 1569, 1554, 1533, 1500 (NH, C=C, C=N), 1234–1130 (CF) cm^–1^; ^1^H NMR (CDCl_3_, 500 MHz) δ 7.18–7.21, 7.29–7.32, 7.44–7.49, 7.93–7.95 (8H, all m, Ph and C_6_H_3_), 13.76 (1H, s, NNH); ^13^C NMR (CDCl_3_, 125 MHz) δ 118.18, 119.26 (q, *J* = 271.4 Hz, CF_3_), 119.38, 125.05, 126.56, 127.97, 128.00, 129.13, 129.83, 134.23, 137.06, 138.47 (q, *J* = 39.5 Hz, C–CF_3_), 156.53; ^19^F NMR (CDCl_3_, 470 MHz) δ 97.15 (s, CF_3_); Anal. calcd. for C_16_H_9_Cl_2_F_3_N_4_. C, 47.90; H, 2.26; N, 13.97. Found: C, 47.72; H, 2.39; N, 13.63.

*(4Z)-4-[2-(4-Bromophenyl)hydrazinylidene]-2-phenyl-5-(trifluoromethyl)-2,4-dihydro-3H-pyrazol-3-one* (**5k**). Yield 1.02 g (83% *method B*); orange powder; mp 185–186 °C; IR ν 3298, 3099, 3071 (NH), 1653 (C=O), 1538, 1478 (NH, C=C, C=N), 1279–1132 (CF) cm^–1^; ^1^H NMR (CDCl_3_, 500 MHz) δ 7.29–7.32, 7.38–7.40, 7.45–7.49, 7.58–7.60 (9H, all m, Ph and C_6_H_4_), 13.93 (1H, s, NNH); ^13^C NMR (CDCl_3_, 125 MHz) δ 118.18, 119.31, 119.43 (q, *J* = 271.3 Hz, CF_3_), 120.71, 123.78, 126.55, 129.15, 133.01, 137.18, 138.21 (q, *J* = 39.3 Hz, C–CF_3_), 139.39, 157.02; ^19^F NMR (CDCl_3_, 470 MHz) δ 97.51 (s, CF_3_); Anal. calcd. for C_16_H_10_BrF_3_N_4_O. C, 46.74; H, 2.45; N, 13.63. Found: C, 46.94; H, 2.53; N, 13.54.

*(4Z)-4-[2-(4-Ethoxycarbonyl)hydrazinylidene]-2-phenyl-5-(trifluoromethyl)-2,4-dihydro-3H-pyrazol-3-one* (**5l**). Yield 0.88 g (73% *method B*); yellow powder; mp 160–162 °C; IR ν 3070, 2991, 2961, 2941 (NH), 1668 (C=O), 1591, 1554, 1536, 1499 (NH, C=C, C=N), 1282–1128 (CF) cm^–1^; ^1^H NMR (CDCl_3_, 400 MHz) δ 1.42 (3H, t, *J* = 7.1 Hz, OCH_2_CH_3_); 4.40 (2H, q, *J* = 7.1 Hz, OCH_2_CH_3_), 7.29–7.33, 7.46–7.50, 7.54–7.56, 7.92–7.94, 8.14–8.16 (9H, all m, Ph and C_6_H_4_), 13.91 (1H, s, NNH); ^13^C NMR (CDCl_3_, 125 MHz) δ 14.29, 61.25, 116.25, 119.25, 119.34 (q, *J* = 271.3 Hz, CF_3_), 124.58, 126.59, 128.92, 129.14, 131.45, 137.06, 138.29 (q, *J* = 39.4 Hz, C–CF_3_), 143.57, 156.79, 165.52; ^19^F NMR (CDCl_3_, 376 MHz) δ 97.45 (s, CF_3_); Anal. calcd. for C_19_H_15_F_3_N_4_O_3_. C, 56.44; H, 3.74; N, 13.86. Found: C, 56.48; H, 3.76; N, 13.76.

(*4Z)-4-(2-[4-(Methylsulfonyl)phenyl]hydrazinylidene)-2-phenyl-5-(trifluoromethyl)-2,4-dihydro-3H-pyrazol-3-one* (**5m**). Yield 1.01 g (82% *method B*); orange powder; mp 230–231 °C; IR ν 3127, 3094, 3067, 2927 (NH), 1667 (C=O), 1558, 1534, 1501, 1485 (NH, C=C, C=N), 1270–1127 (CF) cm^–1^; ^1^H NMR (CDCl_3_, 500 MHz) δ 3.09 (3H, s, SO_2_CH_3_); 7.31–7.34, 7.47–7.50, 7.65–7.67, 7.91–7.92, 8.03–8.05 (9H, all m, Ph and C_6_H_4_), 13.86 (1H, s, NNH); ^13^C NMR (CDCl_3_, 125 MHz) δ 16.94, 116.94, 119.19 (q, *J* = 271.2 Hz, CF_3_), 119.30, 125.53, 126.81, 129.21, 129.58, 136.87, 138.23, 138.34 (q, *J* = 39.5 Hz, C–CF_3_), 144.45, 156.66; ^19^F NMR (CDCl_3_, 470 MHz) δ 97.39 (s, CF_3_); Anal. calcd. for C_17_H_13_F_3_N_4_O_3_S. C, 49.76; H, 3.19; N, 13.65; S, 7.81. Found: C, 49.58; H, 3.20; N, 13.54; S, 7.63.

*(4Z)-4-(2-[4-(Aminosulfonyl)phenyl]hydrazinylidene)-2-phenyl-5-(trifluoromethyl)-2,4-dihydro-3H-pyrazol-3-one* (**5n**). Yield 0.92 g (78% *method B*); orange powder; mp 255–256 °C. IR ν 3390, 3318, 3241 (NH,), 1675 (C=O), 1542, 1486, 1448 (NH, C=C, C=N), 1160–1131 (CF) cm^–1^; ^1^H NMR (DMSO-*d*_6_, 500 MHz) δ 7.35 (2H, s, NH_2_), 7.24–7.25, 7.45–7.48, 7.72–7.74, 7.86–7.87, 7.95–7.96 (9H, all m, Ph and C_6_H_4_), 11.91 (1H, br. s, NNH); ^13^C NMR (DMSO-*d*_6_, 125 MHz) δ: 118.49, 118.97 (2C), 120.43 (q, *J* = 271.8 Hz, CF_3_), 123.01, 125.46, 127.08, 128.98 (2C), 138.13 (br. s), 141.37, 171.98; ^19^F NMR (DMSO-*d*_6_, 470 MHz) δ 100.00 (s, CF_3_); Anal. calcd. for C_16_H_12_F_3_N_5_O_3_S. C, 46.72; H, 2.91; N, 17.03; S, 7.79. Found, %: C, 46.98; H, 2.72; N, 17.23; S, 7.92.

*(4Z)-4-(2-[4-(Sulfo)phenyl]hydrazinylidene)-2-phenyl-5-(trifluoromethyl)-2,4-dihydro-3H-pyrazol-3-one* (**5o**). Yield 0.80 g (67% *method B*); orange powder; mp 360–361 °C; IR ν 3213, 3177 (NH), 1660 (C=O), 1539, 1487, 1450 (NH, C=C, C=N), 1225–1123 (CF) cm^–1^; ^1^H NMR (DMSO-*d*_6_, 400 MHz) δ 7.31–7.35, 7.50–7.54, 7.62–7.64, 7.69–7.72, 7.87–7.89 (9H, all m, Ph and C_6_H_4_), the signals of OH and NNH were not observed due to the deuteroexchange; ^13^C NMR (DMSO-*d*_6_, 125 MHz) δ 116.79, 118.69 (2C), 119.77 (q, *J* 270.3 Hz, CF_3_), 122.91, 126.15, 127.06, 129.18, 137.30, 137.81 (q, *J* 42.8 Hz, C–CF_3_), 146.92, 155.56; ^19^F NMR (DMSO-*d*_6_, 376 MHz) δ 98.65 (s, CF_3_); Anal. calcd. for C_16_H_11_F_3_N_4_O_4_S. C, 46.61; H, 2.69; N, 13.59; S, 7.78. Found: C, 46.48; H, 2.44; N, 13.59; S, 7.65.

*(4Z)-2-[4-(Aminosulfonyl)phenyl]-4-[2-(4-methylphenyl)hydrazinylidene]-5-(trifluoromethyl)-2,4-dihydro-3H-pyrazol-3-one* (**5p**). Yield 0.89 g (70% *method B*); yellow powder; mp 277–278 °C; IR ν 3340, 3258 (NH), 1669 (C=O), 1537, 1494 (NH, C=C, C=N), 1239–1141 (CF) cm^–1^; ^1^H NMR (DMSO-*d*_6_, 400 MHz) δ 2.34 (3H, s, CH_3_), 7.32 (2H, d, *J* = 8.5 Hz, C_6_H_4_–CH_3_), 7.41 (2H, s, NH_2_), 7.60 (2H, d, *J* = 8.5 Hz, C_6_H_4_–CH_3_), 7.96 (2H, d, *J* = 8.1 Hz, C_6_H_4_–SO_2_NH_2_), 8.11 (2H, d, *J* = 8.1 Hz, C_6_H_4_–SO_2_NH_2_); 13.58 (1H, br. s, NNH); ^13^C NMR (DMSO-*d*_6_, 125 MHz) δ 20.66, 117.60, 118.28, 119.76 (q, *J* = 270.7 Hz, CF_3_), 121.80, 127.00, 130.16, 137.27, 137.87 (q, *J* = 37.5 Hz, C–CF_3_), 139.62, 139.88, 140.89, 155.88; ^19^F NMR (DMSO-*d*_6_, 376 MHz) δ 99.42 (s, CF_3_); Anal. calcd. for C_17_H_14_F_3_N_5_O_3_S. C, 48.00; H, 3.32; N, 16.46; S, 7.54. Found: C, 48.22; H, 3.42; N, 16.53; S, 7.62.

*(4Z)-2-[4-(Methylsulfonyl)phenyl]-4-[2-(4-methylphenyl)hydrazinylidene]-5-(trifluoromethyl)-2,4-dihydro-3H-pyrazol-3-one* (**5q**). Yield 0.77 g (61% *method B*); yellow powder; mp 275–276 °C. IR ν 3125, 3034, 2934 (NH), 1674 (C=O), 1556, 1536 (NH, C=C, C=N), 1236–1138 (CF) cm^−1^; ^1^H NMR (DMSO-*d*_6_, 500 MHz) δ: 2.35, 3.25 (6H, both s, 2 CH_3_), 7.33 (2H, d, *J* = 8.8 Hz, C_6_H_4_–Me), 7.61 (2H, d, *J* = 8.8 Hz, C_6_H_4_–Me), 8.07 (2H, d, *J* = 8.4 Hz, C_6_H_4_–SO_2_Me), 8.20 (2H, d, *J* = 8.4 Hz, C_6_H_4_–SO_2_Me), the signals of NNH was not observed due to the deuteroexchange; ^19^F NMR (DMSO-*d*_6_, 470 MHz) δ: 99.30 (s, CF_3_); Anal. calcd. for C_18_H_15_F_3_N_4_O_3_S. C, 50.94; H, 3.56; N, 13.20; S, 7.55. Found: C, 50.86; H, 3.59; N, 13.13; S, 7.68.

(*4Z)-2-(4-Methylphenyl)-4-[2-(4-methylphenyl)hydrazinylidene]-5-(trifluoromethyl)-2,4-dihydro-3H-pyrazol-3-one* (**5r**). Yield 0.56 g (52% *method C*); orange powder; mp 184–185 °C; IR ν: 3031, 2930 (NH), 1652 (C=O), 1553, 1532, 1490 (NH, C=C, C=N), 1242–1134 (CF) cm^–1^; ^1^H NMR (CDCl_3_, 400 MHz) δ 2.38, 2.39 (6H, both s, CH_3_), 7.25–7.26, 7.26–7.27, 7.40–7.42, 7.78–7.81 (8H, all m, 2 C_6_H_4_), 14.05 (1H, br. s, NNH); ^13^C NMR (CDCl_3_, 125 MHz) δ 21.00, 21.13, 116.77, 119.34, 119.62 (q, *J* = 271.0 Hz, CF_3_), 122.80, 129.58, 130.43, 134.95, 136.16, 137.87, 137.88 (q, *J* = 39.1 Hz, C–CF_3_), 138.07, 156.98; ^19^F NMR (CDCl_3_, 376 MHz) δ 97.66 (s, CF_3_); Anal. calcd. for C_18_H_15_F_3_N_4_O. C, 60.00; H, 4.20; N, 15.55. Found: C, 60.15; H, 4.15; N, 15.63.

*(4Z)-4-[2-(4-Methylphenyl)hydrazinylidene]-5-(pentafluoroethyl)-2,4-dihydro-3H-pyrazol-3-one* (**6a**). Yield 0.56 g (59% *method A*); yellow powder; mp 174–175 °C; IR ν 3271, 3054, 2928 (NH), 1666 (C=O), 1546, 1519, 1493 (NH, C=C, C=N), 1217–1154 (CF) cm^–1^; ^1^H NMR (CDCl_3_, 500 MHz) δ 2.39 (3H, s, CH_3_), 7.25 (2H, d, *J* = 8.5 Hz, C_6_H_4_), 7.39 (2H, d, *J* = 8.5 Hz, C_6_H_4_), 9.36 (1H, s, NH), 13.88 (1H, br. s, NNH); ^13^C NMR (CDCl_3_, 125 MHz) δ 21.15, 109.37 (tq, *J* = 253.6, 40.2 Hz, CF_2_), 116.88, 118.78 (qt, *J* = 286.8, 36.6 Hz, CF_3_), 122.61, 130.47, 137.78 (t, *J* = 40.2, C—C_2_F_5_), 138.03, 138.11, 159.97. ^19^F NMR (CDCl_3_, 470MHz) δ 46.75 (2F, q, *J* = 2.2 Hz, CF_2_), 78.29 (3F, t, *J* = 2.2 Hz, CF_3_). Anal. calcd. for C_12_H_9_F_5_N_4_O. C, 45.01; H, 2.83; N, 17.50. Found: C, 45.25; H, 2.65; N, 17.59.

*(4Z)-4-[2-(4-Methylphenyl)hydrazinylidene]-5-(pentafluoroethyl)-2-phenyl-2,4-dihydro-3H-pyrazol-3-one* (**6b**). Yield 0.95 g (80% *method B*); yellow powder; mp 128–129 °C; IR ν 3108, 3049, 2924, 2860 (NH), 1660 (C=O), 1551, 1523, 1491 (NH, C=C, C=N), 1231–1115 (CF) cm^–1^; ^1^H NMR (CDCl_3_, 500 MHz) δ 2.40 (3H, s, CH_3_), 7.26–7.31, 7.40–7.48, 7.94–7.95 (9H, all m, C_6_H_4_ and Ph), 14.43 (1H, br. s, NNH). ^13^C NMR (CDCl_3_, 125 MHz) δ 21.17, 109.22 (tq, *J* = 254.4, 40.1 Hz, CF_2_), 116.88, 119.81 (qt, *J* = 286.8, 36.6 Hz, CF_3_), 119.67, 123.48, 126.40, 129.08, 130.49, 137.10 (t, *J* = 28.7, C—C_2_F_5_), 137.39, 138.06, 138.10, 157.26; ^19^F NMR (CDCl_3_, 470 MHz) δ 46.96 (2F, q, *J* = 2.6 Hz, CF_2_); 48.46 (3F, t, *J* = 2.6 Hz, CF_3_); Anal. calcd. for C_18_H_13_F_5_N_4_O. C, 54.55; H, 3.31; N, 14.14. Found: C, 54.85; H, 3.60; N, 14.24.

*(4Z)-4-[2-(2-Chlorophenyl)hydrazinylidene]-5-(pentafluoroethyl)-2-phenyl-2,4-dihydro-3H-pyrazol-3-one* (**6c**). Yield 0.88 g (71% *method B*); orange powder; mp 179–180 °C; IR ν 3319, 3074 (NH), 1664 (C=O), 1551, 1522, 1499 (NH, C=C, C=N), 1218–1117 (CF) cm^–1^; ^1^H NMR (CDCl_3_, 500 MHz) δ 7.22–7.25, 7.29–7.32, 7.40–7.43, 7.46–7.49, 7.88–7.90, 7.95–7.96 (9H, all m, C_6_H_4_ and Ph); 14.23 (1H, s, NNH); ^13^C NMR (CDCl_3_, 125 MHz) δ 109.98 (tq, *J* = 254.6, 40.3, CF_2_), 116.92, 118.50 (qt, *J* = 286.8, 36.4 Hz, CF_3_), 119.64, 123.00, 125.76, 126.59, 127.60, 128.47, 129.14, 130.00, 137.15 (t, *J* = 28.9 Hz, C—C_2_F_5_), 137.17, 137.23, 156.90; ^19^F NMR (CDCl_3_, 470 MHz) δ 46.83 (2F, q, *J* = 2.5 Hz, CF_2_), 78.50 (3F, t, *J* = 2.6 Hz, CF_3_); Anal. calcd. for C_17_H_10_ClF_5_N_4_O. C, 49.00; H, 2.42; N, 13.40. Found: C, 49.01; H, 2.41; N, 13.17.

(*4Z)-4-[2-(3-Chlorophenyl)hydrazinylidene]-5-(pentafluoroethyl)-2-phenyl-2,4-dihydro-3H-pyrazol-3-one* (**6d**). Yield 1.17 g (94% *method B*); orange powder; mp 130–131 °C; IR ν: 3072 (NH), 1668 (C=O), 1548, 1494 (NH, C=C, C=N), 1275–1058 (CF) cm^–1^; ^1^H NMR (CDCl_3_, 500 MHz) δ 7.28–7.41, 7.46–7.49, 7.52–7.53, 7.92–7.94 (9H, all m, C_6_H_4_ and Ph), 13.96 (1H, s, NNH); ^13^C NMR (CDCl_3_, 125 MHz) δ 109.93 (tq, *J* = 254.7, 40.3 Hz, CF_2_), 115.05, 118.47 (qt, *J* = 286.8, 36.4 Hz, CF_3_), 119.29, 124.85, 126.63, 126.59, 127.34, 129.14, 130.91, 136.04, 137.15, 137.28 (t, *J* = 29.0 Hz, C—C_2_F_5_), 141.53, 157.04; ^19^F NMR (CDCl_3_, 470MHz) δ 46.91 (2F, q, *J* = 2.5 Hz, CF_2_), 78.52 (3F, t, *J* = 2.65 Hz, CF_3_); Anal. calcd. for C_17_H_10_ClF_5_N_4_O. C, 49.00; H, 2.42; N, 13.40. Found: C, 48.87; H, 2.40; N, 13.53.

*(4Z)-4-[2-(2,6-Dichlorophenyl)hydrazinylidene]-5-(pentafluoroethyl)-2-phenyl-2,4-dihydro-3H-pyrazol-3-one* (**6e**). Yield 0.86 g (64% *method B*); orange powder; mp 138–140 °C; IR ν 3318, 3068 (NH), 1665, (C=O), 1553, 1500 (NH, C=C, C=N), 1215–1170 (CF) cm^–1^; ^1^H NMR (CDCl_3_, 500 MHz) δ 7.28–7.41, 7.46–7.49, 7.52–7.53, 7.92–7.94 (9H, all m, C_6_H_4_ and Ph), 13.96 (1H, s, NNH). ^13^C NMR (CDCl_3_, 125 MHz) δ 109.82 (tq, *J* = 254.8, 40.2 Hz, CF_2_), 119.41 (2C), 119.57 (qt, *J* = 286.8, 36.4 Hz, CF_3_), 125.83, 126.64, 128.03, 129.15, 129.86, 134.30, 137.09, 137.55 (t, *J* = 29.2 Hz, C—C_2_F_5_), 156.68. ^19^F NMR (CDCl_3_, 470 MHz) δ 46.79 (2F, q, *J* = 2.5 Hz, CF_2_), 78.56 (3F, t, *J* = 2.5 Hz, CF_3_). Anal. calcd. for C_17_H_9_Cl_2_F_5_N_4_O. C, 45.26; H, 2.01; N, 12.42. Found: C, 45.12; H, 2.05; N, 12.28.

*(4Z)-4-[2-(2,4-Dichlorophenyl)hydrazinylidene]-5-(pentafluoroethyl)-2-phenyl-2,4-dihydro-3H-pyrazol-3-one* (**6f**). Yield 0.99 g (73% *method B*); orange powder; mp 173–174 °C; IR ν 3318, 3070 (NH), 1663 (C=O), 1551, 1499, 1455 (NH, C=C, C=N), 1229–1101 (CF) cm^–1^; ^1^H NMR (CDCl_3_, 500 MHz) δ 7.29–7.32, 7.37–7.39, 7.46–7.49, 7.81–7.82, 7.93–7.95 (9H, all m, C_6_H_4_ and Ph), 14.18 (1H, s, NNH). ^13^C NMR (CDCl_3_, 125 MHz) δ 109.91 (tq, *J* = 254.7, 40.3 Hz, CF_2_), 117.65, 118.46 (qt, *J* = 286.7, 36.4 Hz, CF_3_), 119.35, 123.29, 126.14, 126.70, 128.91, 129.17, 129.72, 132.55, 136.07, 137.05, 137.10 (t, *J* = 29.0 Hz, C—C_2_F_5_), 156.85; ^19^F NMR (CDCl_3_, 470 MHz) δ 46.84 (2F, q, *J* = 2.5 Hz, CF_2_); 78.52 (3F, t, *J* = 2.5 Hz, CF_3_); Anal. calcd. for C_17_H_9_Cl_2_F_5_N_4_O. C, 45.26; H, 2.01; N, 12.42. Found: C, 45.44; H, 1.93; N, 12.37.

*(4Z)-4-[2-(4-Ethoxycarbonylphenyl)hydrazinylidene]-5-(pentafluoroethyl)-2-phenyl-2,4-dihydro-3H-pyrazol-3-one* (**6g**). Yield 1.13 g (83% *method B*); orange powder; mp 145–146 °C; IR ν 3070, 3049, 2986, 2910 (NH), 1703 (C=O), 1549, 1515, 1491 (NH def., C=C, C=N), 1288–1060 (CF) cm^–1^; ^1^H NMR (CDCl_3_, 500 MHz) δ 1.41 (3H, t, *J* = 7.1 Hz, CH_2_CH_3_), 4.40 (2H, q, *J* = 7.1 Hz, CH_2_CH_3_), 7.26–7.33, 7.46–7.50, 7.53–7.55, 7.92–7.94, 8.14–8.16 (9H, all m, C_6_H_4_ and Ph), 13.99 (1H, br. s, NNH); ^13^C NMR (CDCl_3_, 125 MHz) δ 14.28, 61.24, 109.90 (tq, *J* = 254.5, 40.4 Hz, CF_2_), 116.30, 119.25, 119.59 (qt, *J* = 286.7, 36.5 Hz, CF_3_), 125.33, 126.65, 129.02, 129.14, 131.48, 137.09, 137.34 (t, *J* = 29.0 Hz, C—C_2_F_5_), 143.66, 156.91, 165.51; ^19^F NMR (CDCl_3_, 470 MHz) δ 46.87 (2F, q, *J* = 2.5 Hz, CF_2_), 78.53(3F, t, *J* = 2.5 Hz, CF_3_); Anal. calcd. for C_20_H_15_F_5_N_4_O_3_. C, 52.87; H, 3.33; N, 12.33. Found: C, 52.76; H, 3.37; N, 12.17.

*(4Z)-4-(2-[4-(Methylsulfonyl)phenyl]hydrazinylidene)-5-(pentafluoroethyl)-2-phenyl-2,4-dihydro-3H-pyrazol-3-one* (**6h**). Yield 0.93 g (68% *method B*); yellow powder; mp 187–190 °C; IR ν 3074, 3019, 2935 (NH), 1666 (C=O), 1554, 1528, 1492 (NH, C=C, C=N), 1212–1066 (CF) cm^–1^; ^1^H NMR (CDCl_3_, 500 MHz) δ 3.09 (3H, s, SO_2_CH_3_), 7.26–7.34, 7.47–7.51, 7.64–7.66, 7.91–7.93, 8.04–8.05 (9H, all m, C_6_H_4_ and Ph), 13.94 (1H, br. s, NNH); ^13^C NMR (CDCl_3_, 125 MHz) δ 44.65, 109.84 (tq, *J* = 254.9, 40.4 Hz, CF_2_), 117.03, 119.33, 119.81 (qt, *J* = 286.7, 36.2 Hz, CF_3_), 126.28, 126.91, 129.25, 129.63, 136.94, 137.45 (t, *J* = 29.1 Hz, C—C_2_F_5_), 138.40, 144.56, 156.81; ^19^F NMR (CDCl_3_, 470 MHz) δ 46.81 (2F, q, *J* = 2.5 Hz, CF_2_), 78.56 (3F, t, *J* = 2.5 Hz, CF_3_); Anal. calcd. for C_18_H_13_F_5_N_4_O_3_S. C, 46.96; H, 2.85; N, 12.17; S, 6.96. Found: C, 46.10; H, 2.71; N, 12.33; S, 6.93.

*(4Z)-4-(2-[4-Methylphenyl]hydrazinylidene)-2-(4-methylphenyl)-5-(pentafluoroethyl)-2,4-dihydro-3H-pyrazol-3-one* (**6i**). Yield 0.59 g (48% *method C*); orange powder; mp 165–166 °C;^1^H NMR (CDCl_3_, 400 MHz) δ 2.39, 2.40 (6H, both s, CH_3_), 7.25–7.26, 7.26–7.27, 7.39–7.41, 7.79–7.81 (8H, all m, 2 C_6_H_4_), 14.14 (1H, br. s, NNH); ^19^F NMR (CDCl_3_, 376 MHz) δ 46.98 (2F, q, *J* = 2.6 Hz, CF_2_), 78.44 (3F, t, *J* = 2.6 Hz, CF_3_); Anal. calcd. for C_19_H_15_F_5_N_4_O. C, 55.61; H, 3.68; N, 13.65. Found: C, 55.65; H, 3.79; N, 13.48.

*(4Z)-5-(Heptafluoropropyl)-4-[2-(4-methylphenyl)hydrazinylidene]-2,4-dihydro-3H-pyrazol-3-one* (**7a**). Yield 0.72 g (65% *method B*); orange powder; mp 165–167 °C; IR ν 3273, 3048 (NH), 1662 (C=O), 1598, 1543, 1511, 1487 (NH, C=C, C=N), 1212–1182 (CF) cm^–1^; ^1^H NMR (CDCl_3_, 400 MHz) δ 2.39 (3H, s, CH_3_), 7.26 (2H, d, *J* = 8.5 Hz, C_6_H_4_), 7.39 (2H, d, *J* = 8.5 Hz, C_6_H_4_), 9.63 (1H, s, NH), 13.91 (1H, br. s, NNH); ^13^C NMR (CDCl_3_, 125 MHz) δ 21.11, 108.65 (tq, *J* = 266.4, 37.8 Hz, CF_2_–CF_2_–CF_3_), 111.78 (tt, *J* = 256.3, 32.2 Hz, CF_2_–CF_2_–CF_3_), 116.93, 117.87 (qt, *J* = 287.6, 33.6 Hz, CF_2_–CF_2_–CF_3_), 122.89, 130.53, 138.01 (t, *J* = 28.2 Hz, C–C_3_F_7_), 138.14, 138.17, 160.09; ^19^F NMR (CDCl_3_, 376 MHz) δ 35.15–35.23, 48.55–48.62 (4F, both m, 2 CF_2_), 81.55 (3F, t, *J* = 9.3 Hz, CF_3_); Anal. calcd. for C_13_H_9_F_7_N_4_O. C, 42.17; H, 2.45; N, 15.13. Found: C, 42.02; H, 2.59; N, 15.01.

*(4Z)-5-(Heptafluoropropyl)-4-[2-(4-methylphenyl)hydrazinylidene]-2-phenyl-2,4-dihydro-3H-pyrazol-3-one* (**7b**). Yield 0.99 g (75% *method B*); orange powder; mp 132–133 °C; IR ν 3113, 3035, 2928, 2860 (NH), 1666 (C=O), 1546, 1516, 1489 (NH, C=C, C=N), 1212–1116 (CF) cm^–1^; ^1^H NMR (CDCl_3_, 500 MHz) δ 2.40 (3H, s, CH_3_), 7.26–7.31, 7.39–7.41, 7.45–7.49, 7.94–7.96 (9H, all m, C_6_H_4_ and Ph), 14.15 (1H, br. s, NNH); ^13^C NMR (CDCl_3_, 125 MHz) δ 21.16, 108.66 (tq, *J* = 266.5, 37.9 Hz, CF_2_–CF_2_–CF_3_), 111.94 (tt, *J* = 256.7, 32.4 Hz, CF_2_–CF_2_–CF_3_), 116.89, 117.89 (qt, *J* = 287.4, 34.0 Hz, CF_2_–CF_2_–CF_3_), 119.40, 123.71, 126.43, 129.09, 130.51, 137.16 (t, *J* = 28.4 Hz, C–C_3_F_7_), 137.44, 138.07, 138.16, 157.33; ^19^F NMR (CDCl_3_, 470 MHz) δ 35.48–35.50, 48.86–48.91 (4F, both m, 2 CF_2_), 81.67 (3F, t, *J* = 9.3 Hz, CF_3_); Anal. calcd. for C_19_H_13_F_7_N_4_O. C, 51.13; H, 2.94; N, 12.55. Found: C, 51.25; H, 2.72; N, 12.58.

*(4Z)-5-(Nonafluorobutyl)-4-[2-(4-methylphenyl)hydrazinylidene]-2,4-dihydro-3H-pyrazol-3-one* (**8a**). Yield 0.98 g (78% *method B*); yellow powder; mp 175–176 °C; IR ν 3278, 3047, 2928 (NH), 1666 (C=O), 1546, 1514, 1491 (NH, C=C, C=N), 1292–1199 (CF) cm^–1^; ^1^H NMR (CDCl_3_, 400 MHz) δ 2.39 (3H, s, CH_3_), 7.25 (2H, d, *J* = 8.5 Hz, C_6_H_4_), 7.39 (2H, d, *J* = 8.5 Hz, C_6_H_4_), 9.75 (1H, s, NH), 13.91 (1H, br. s, NNH); ^13^C NMR (CDCl_3_, 125 MHz) δ 21.15,108.40–118.85 (m, C_4_F_9_), 116.87,122.75, 130.49, 138.05, 138.12, 138.14 (unsolv. t, C—C_4_F_9_), 159.78; ^19^F NMR (CDCl_3_, 376 MHz) δ 35.93–36.04, 38.78–38.85, 49.36–49.43 (6F, all m, 4CF_2_); 80.86 (3F, tt, *J* = 9.7, 2.4 Hz, CF_3_); Anal. calcd. for C_14_H_9_F_9_N_4_O. C, 40.01; H, 2.16; N, 13.33. Found: C, 40.21; H, 2.17; N, 13.38.

*(4Z)-5-(Nonafluorobutyl)-4-[2-(4-methylphenyl)hydrazinylidene]-2-phenyl-2,4-dihydro-3H-pyrazol-3-one* (**8b**). Yield 1.41 g (95% *method B*); yellow powder; mp 142–144 °C; IR ν 3116, 3083, 3047, 2929, 2867 (NH), 1666 (C=O), 1552, 1517, 1490 (NH, C=C, C=N), 1227–1179 (CF) cm^–1^; ^1^H NMR (CDCl_3_, 400 MHz) δ 2.40 (3H, s, CH_3_), 7.26–7.29, 7.39–7.41, 7.45–7.49, 7.94–7.96 (9H, all m, C_6_H_4_ and Ph), 14.16 (1H, br. s, NNH); ^13^C NMR (CDCl_3_, 125 MHz) δ 21.18, 106.63–120.95 (m, C_4_F_9_), 116.92, 119.42, 123.76, 126.46, 129.12, 130.54, 137.30 (t, *J* = 28.4 Hz, C–C_4_F_9_), 137.48, 138.10, 138.20, 157.37; ^19^F NMR (CDCl_3_, 376 MHz) δ 36.09–36.17, 39.06–39.13, 49.65–49.72 (6F, all m, 3 CF_2_); 80.88 (3F, tt, *J* = 9.7, 2.5 Hz, CF_3_); Anal. calcd. for C_20_H_13_F_9_N_4_O. C, 48.40; H, 2.64; N, 11.29. Found: C, 48.45; H, 2.68; N, 11.19.

*(4Z)-5-Methyl-4-[2-(4-methylphenyl)hydrazinylidene]-2,4-dihydro-3H-pyrazol-3-one* (**9a**). Yield 0.40 g (61% *method A*); orange powder; mp 197–198 °C; IR ν 3290, 3160, 3027 (NH), 1666 (C=O), 1593, 1545, 1492 (NH, C=C, C=N), 1247–1174 (CF) cm^–1^; ^1^H NMR (CDCl_3_, 400 MHz) δ 2.27 (3H, s, CH_3_), 2.36 (3H, s, CH_3_ Ar), 7.21 (2H, d, *J* = 8.2 Hz, C_6_H_4_), 7.32 (2H, d, *J* = 8.5 Hz, C_6_H_4_), 8.79 (1H, s, NH), 13.40 (1H, br. s, NNH). ^13^C NMR (CDCl_3_, 125 MHz) δ 11.80, 20.96, 115.73, 126.98, 130.12, 135.71, 138.85, 148.58, 160.88; Anal. calcd. for C_11_H_12_N_4_O. C, 61.10; H, 5.59; N, 25.91. Found: C, 61.25; H, 5.68; N, 25.71.

(*4Z)-5-Methyl-4-[2-(4-methylphenyl)hydrazinylidene]-2-phenyl-2,4-dihydro-3H-pyrazol-3-one* (**9b**). Yield 0.55 g (63% *method A*); orange powder; mp 137–138 °C; IR ν 3292, 3064, 2924, 2863 (NH), 1652 (C=O), 1593, 1540, 1495 (NH, C=C, C=N), 1229–1112 (CF) cm^–1^; ^1^H NMR (CDCl_3_, 500 MHz) δ 2.37 (6H, s, 2 CH_3_), 7.19–7.23, 7.33–7.35, 7.41–7.44, 7.95–7.96 (9H, all m, C_6_H_4_ and Ph), 13.63 (1H, s, NNH). ^13^C NMR (CDCl_3_, 125 MHz) δ 11.78, 21.01, 115.78, 118.54, 125.05, 127.92, 128.89, 130.21, 135.89, 138.11, 138.84, 148.48, 157.87. Anal. calcd. for C_17_H_16_N_4_O. C, 69.85; H, 5.52; N, 19.17. Found: C, 69.95; H, 5.56; N, 19.19.

#### 3.1.2. XRD Experiments

The X-ray studies were performed on an Xcalibur 3 CCD (Oxford Diffraction Ltd., Abingdon, UK) diffractometer with a graphite monochromator, λ(MoKα) 0.71073 Å radiation, T 295(2) K. An empirical absorption correction was applied. Using Olex2 [107], the structure was solved with the Superflip [108] structure solution program using Charge Flipping and refined with the ShelXL [109] refinement package using Least Squares minimization. All non-hydrogen atoms were refined in the anisotropic approximation; H-atoms at the C–H bonds were refined in the “rider” model with dependent displacement parameters. An empirical absorption correction was carried out through spherical harmonics, implemented in the SCALE3 ABSPACK scaling algorithm by the program “CrysAlisPro” (Rigaku Oxford Diffraction).

The suitable single crystals of compound **5b** were obtained by slow crystallization from acetonitrile. Main crystallographic data for **5b**: C_11_H_9_F_3_N_4_O, *M* 270.22, tetragonal, *a* 19.0081(9), *c* 7.0383(8) Å, *V* 2543.0(3) Å^3^, space group I4/m, *Z* 8, μ(Mo Kα) 0.125 mm^–1^, 147 refinement parameters, 8881 reflections measured, 1938 unique (*R*_int_ = 0.0351), which were used in all calculations. The final *wR*_2_ was 0.2122 (all data) and *R*_1_ was 0.0597 (>2σ(I)). CCDC 2,111,746 contains the supplementary crystallographic data for this compound.

The suitable single crystals of compound **6c** were obtained by slow crystallization from CDCl_3_. Main crystallographic data for **6c**: C_20_H_15_F_5_N_4_O_3_, *M* 454.36, triclinic, *a* 9.6333(4), *b* 12.4904(5), *c* 18.5502(9) Å, *α* 72.157(4), *β* 80.610(4), *γ* 77.417(4)°, *V* 2062.44(16) Å^3^, space group P$\bar{1}$, *Z* 4, μ(Mo Kα) 0.130 mm^–1^, 650 refinement parameters, 16,250 reflections measured, 10,915 unique (*R*_int_ = 0.0338), which were used in all calculations. The final *wR*_2_ was 0.2528 (all data) and *R*_1_ was 0.0715 (>2σ(I)). CCDC 2,112,124 contains the supplementary crystallographic data for this compound.

The suitable single crystals of compound **6g** were obtained by slow crystallization from CDCl_3_. Main crystallographic data for **6g**: C_17_H_10_ClF_5_N_4_O, *M* 416.74, monoclinic, *a* 5.65826(16), *b* 19.7725(7), *c* 15.3673(5) Å, *β* 97.418(3)°, *V* 1704.87(9) Å^3^, space group P2_1_/c, *Z* 4, μ(Mo Kα) = 0.293 mm^–1^, 257 refinement parameters, 7421 reflections measured, 4192 unique (*R*_int_ = 0.0286), which were used in all calculations. The final *wR*_2_ was 0.1399 (all data) and *R*_1_ was 0.0479 (>2σ(I)). CCDC 2,111,751 contains the supplementary crystallographic data for this compound.

### 3.2. Biological Studies

#### 3.2.1. Inhibition of Porcine Liver CES, Human AChE and Equine BChE

The following enzymes were purchased from Sigma-Aldrich (Saint Louis, MO, USA): porcine liver carboxylesterase (CES, EC 3.1.1.1), human erythrocyte acetylcholinesterase (AChE, EC 3.1.1.7) and equine serum butyrylcholinesterase (BChE, EC 3.1.1.8). Substrates for CES, AChE and BChE were 4-nitrophenyl acetate (4-NPA), acetylthiocholine iodide (ATCh) and butyrylthiocholine iodide (BTCh), respectively, and the colorimetric reagent for the AChE and BChE assays was 5,5’-dithio-bis-(2-nitrobenzoic acid) (DTNB) (Sigma-Aldrich). All the kinetic experiments were performed under standard conditions, according to the protocol of IPAC RAS for a reversible inhibitors study.

The CES (porcine liver) activity was assessed colorimetrically in 0.1 M K/Na phosphate buffer pH 8.0, 25 °C, by measuring the absorbance of 4-nitrophenol at 405 nm [110]. Final enzyme and substrate (4-NPA) concentrations were 0.02 unit/mL and 1 mM, respectively. Reagent blanks included all constituents except enzyme.

Ellman’s colorimetric assay was used to measure AChE and BChE activity in 0.1 M K/Na phosphate buffer at pH 7.5, 25 °C [111]. Final concentrations of reactants were 0.33 mM DTNB, 0.02 unit/mL of AChE or BChE and 1 mM of substrate (ATCh or BTCh, respectively). Reagent blanks consisted of reaction mixtures without enzyme to assess non-enzymatic hydrolysis of substrates.

Test compounds were dissolved in DMSO. Reaction mixtures contained a final DMSO concentration of 2% (*v/v*). Enzyme inhibition was first assessed at a single concentration of 20 µM for each compound after a 5 min incubation at 25 °C in three separate experiments. The most active compounds (inhibition ≥ 30%) were then selected for determination of the IC_50_ (inhibitor concentration resulting in 50% inhibition of control enzyme activity). Compounds (eight concentrations ranging between 10^–11^ and 10^–4^ M were selected to achieve 20 to 80% inhibition) were incubated with each enzyme for 10 min at 25 °C. Substrate was then added and residual enzyme activity relative to an inhibitor-free control was measured using a Bio-Rad Benchmark Plus microplate reader (Marnes-la-Coquette, France).


*Porcine liver CES inhibition kinetics determination*


To elucidate the inhibition mechanisms for active compounds, porcine liver CES residual activities were determined in the presence of 3 increasing concentrations of a test compound and 5–6 decreasing concentrations of the substrates. Test compounds were incubated with the enzymes at 25 °C for 5 min, followed by the addition of substrates. A parallel control was made for an assay of the rate of hydrolysis of the same concentrations of substrates in solutions with no inhibitor. Linear regression of 1/V versus 1/[S] double-reciprocal (Lineweaver-Burk) plots was used to determine the inhibition constants for the competitive component (*K*_i_) and noncompetitive component (α*K*_i_).

Measurements were performed in a Bio-Rad Benchmark Plus microplate reader. Each experiment was performed in triplicate. Plots, linear regressions and values of IC_50_, were determined using Origin 6.1 for Windows, OriginLab (Northampton, MA, USA). Results were calculated using GraphPad Prism version 6.05 for Windows, GraphPad Software (La Jolla, CA, USA) and presented as mean ± SEM.

#### 3.2.2. ABTS Radical Cation Scavenging Activity Assay

Radical scavenging activity of the compounds was evaluated by the ABTS radical cation (ABTS^•+^) scavenging assay by the ability of the compounds to decolorize the ABTS^•+^ solution [112] with some modifications [87]. Trolox was used as a reference standard. Edaravone was used as the comparison compound. All tested compounds and standards were dissolved in DMSO. ABTS (2,2′-azinobis-(3-ethylbenzothiazoline-6-sulfonic acid)) was purchased from Tokyo Chemical Industry Co. Ltd. (Tokyo, Japan). Potassium persulfate (di-potassium peroxdisulfate), Trolox^®^ (6-hydroxy-2,5,7,8-tetramethychroman-2-carboxylic acid), ascorbic acid, DMSO and HPLC-grade ethanol were obtained from Sigma-Aldrich Chemical Co. Aqueous solutions were prepared using deionized water.

The solution of ABTS^•+^ was produced by mixing 7 mM ABTS aqueous solution with 2.45 mM potassium persulfate aqueous solution in equal quantities and allowing them to react for 12–16 h at room temperature in the dark. Radical scavenging capacity of the compounds was analyzed by mixing 10 μL of compound with 240 μL of ABTS^•+^ working solution in ethanol (100 μM final concentration), and after 1 h of incubation at 30 °C the decrease in absorbance was measured spectrophotometrically at 734 nm using a Bio-Rad xMark microplate UV/VIS spectrophotometer (Bio-Rad, Hercules, CA, USA). The compounds were tested in the concentration range of 5 × 10^–7^–4 × 10^–4^ M. Ethanol blanks were run in each assay. Values were obtained from five replicates of each sample and three independent experiments.

Antioxidant activity was calculated as Trolox equivalent antioxidant capacity (TEAC values), consisting of the ratio between the slopes obtained from the linear correlation for concentrations of the tested compounds and Trolox with absorbance of ABTS radical. For the test compounds, which reduce ABTS^•+^ absorbance by more than 60% at 100 µM, we also determined IC_50_ values as 50% reduction of the ABTS radical. The calculations were carried out using Origin 6.1 for Windows. Results were presented as mean ± SEM, calculated using GraphPad Prism version 6.05 for Windows, GraphPad Software.

#### 3.2.3. Ferric Reducing Antioxidant Power (FRAP) Assay

The FRAP assay measures the ability of anti-oxidants to reduce the ferric 2,4,6-tripyridyl-s-triazine complex [Fe(TPTZ)_2_]^3+^ to the intensely blue ferrous colored complex [Fe(TPTZ)_2_]^2+^ with an absorption maximum at λ = 593 nm [113,114]. The reducing ability of a compound is an indicator of its potential anti-oxidant activity [115]. 2,4,6-Tris(pyridin-2-yl)-1,3,5-triazine (TPTZ), FeCl_3_·6H_2_O, Trolox and DMSO were obtained from Sigma-Aldrich Chemical Co (St. Louis, MO, USA).

The ferric reducing ability of the compounds was determined by the method [114] in microplate adapted version described in [116]. The FRAP reagent was prepared by mixing acetate buffer (0.3 M, pH 3.6), TPTZ (10 mM in 40 mM HCl) and FeCl_3_·6H_2_O (20 mM in distilled water) in a ratio of 10:1:1 immediately before use. Compounds were dissolved in DMSO and tested in the concentration 20 μM. The solvent content in the reaction mixture was 4% (*v/v*). The test compounds (10 μL of 0.5 mM solution) were added to the FRAP reagent solution (240 μL) and mixed thoroughly. The reaction was carried out at 37 °C in the dark; the incubation time was 1 h. The absorbance at 600 nm was monitored spectrophotometrically by a FLUOStar OPTIMA microplate reader (BMG Labtech, Ortenberg, Germany) at 37 °C. Trolox was used as a reference compound. Values were obtained from four replicates of each sample in three independent experiments.

The ferric reducing ability of compounds was expressed as TE units (antioxidant activity in Trolox equivalent), the values calculated as the ratio of the concentrations of Trolox and the test compound resulting in the same effect. The calculations were carried out using Origin 6.1 for Windows. Results were presented as mean ± SEM, calculated using GraphPad Prism version 6.05 for Windows, GraphPad Software.

#### 3.2.4. Oxygen Radical Absorbance Capacity (ORAC) Assay

The ORAC assay was performed according to the known method [117], with minor variations and described by us recently [118].

#### 3.2.5. Antimicrobial Activity

Antimicrobial activity of compounds was determined in relation to control fungi stains *Trichophyton rubrum* RCPF (the Russian Collection of Pathogenic Fungi) F 1408, *Epidermophyton floccosum* RCPF F 1659/17, *Microsporum canis* RCPF F 1643/1585, *Candida albicans* RCPF Y 401/NCTC 885/653 and bacteria—*Neisseria gonorrhoeae* ATCC 49226/NCTC 12700.

Antifungal susceptibility testing of microconidia-forming dermatophytes was performed according to reference EUCAST methods (E.Def 11.0) [119], for yeast-like organisms—E.Def 7.1 and E.Def 7.2. [120,121]. The solvent was DMSO. Final concentrations were in the range: 200 to 0.097 µg/mL. Experimental microplates with dermatophytes were incubated for 7 days, with yeast-like fungi for 24 h at a temperature of 27 °C. In each test, a growth control and a sterility control were used. The criterion for high antimycotic activity of chemical compounds was considered to be MIC < 0.19–6.25 µg/mL; moderate activity—12.5–25 µg/mL.

Antibacterial activity testing was done on the growth medium Difco GC medium base (Becton Dickinson, Franklin Lakes, NJ, USA) supplemented with 1% IsoVitaleX. Dilutions of chemical compounds were carried out in the range of 250–0.12 μg/mL in 24 well sterile plates (Sarstedt, Germany). Daily cultures of control bacterial strains were identified on a VITEK MS analyzer (BioMérieux, Marcy-l’Étoile, France) by MALDI-TOF mass spectrometry. The inoculation dose of *N. gonorrhoeae* inoculum was 10^5^ CFU/mL. Microplates were incubated for 24 h at a temperature of 37 °C, CO_2_—5%; MIC values not exceeding 15 μg/mL were taken as a criterion for the promising chemical compounds.

#### 3.2.6. *In Vitro* Experiments


*Staining*


The suitability of substances for fluorescent imaging of cellular structures was studied on Vero cell culture (African green monkey kidney epithelium) and CHO (Chinese hamster ovary). The culture is supported in DMEM (Gibco, Uoltem, USA) with addition of 10% fetal bovine serum (Gibco, Uoltem, USA) and a mixture of gentamicin and streptomycin (Biolot, Moscow, Russia).

The cells were cultivated into specialized Petri dishes with a glass bottom for staining. Staining was carried out with substances at a concentration of 1 × 10^–4^ M. For that, the solution of substances was preliminarily diluted with DMSO (Biolot, Moscow, Russia) to a concentration of 1 × 10^–3^ M and then with DMEM to the required concentration. Incubation with the substance was carried out for half an hour in an incubator, in the dark at a temperature of 37 °C. After incubation, the cells were washed three times with pure nutrient medium.


*Observation*


The stained cells with the substance were examined using an LSM-710 laser scanning confocal microscope (Carl Zeiss, Oberkochen, Germany). The study was carried out on living cells, without any fixation. Fluorescence was excited by lasers with different wavelengths and registered in lambda mode. This mode gets to collect a full spectrum image by analyzing the intensity of fluorescence in different ranges. The full range of the microscope in this mode is 410–700 nm. Taking images in this mode also makes it possible to obtain the fluorescence spectrum of the test substance in the cell. However, a confocal microscope is not a spectrofluorometer and the resulting spectra can have very poor quality, especially at low fluorescence intensity.


*Cytotoxicity studies*


Cytotoxicity of compounds was evaluated using a culture of human dermal fibroblasts, which were isolated in the Laboratory of Cell Cultures of the Institute of Medical Cell Technologies, Ekaterinburg, Russia and a HeLa tumor cell line obtained at the shared research facility «Vertebrate cell culture collection» of the Institute of Cytology of the Russian Academy of Sciences, St. Petersburg, Russia. Cells were seeded in 96-well plates in the inoculum dose of 2 × 10^5^ cells/mL and cultured for 24 h in DMEM, with 1% glutamine in the presence of 10% fetal bovine serum and gentamicin (50 mg/L) at 37 °C, with humidified atmosphere of 5% CO_2_. Then compounds, which were dissolved in DMSO, were added to the wells at the final concentrations of 10^–7^, 10^–6^, 10^–5^ and 10^–4^ M. The cells were incubated with compounds for 72 h, after which cell viability was assessed using the standard MTT assay [122]. The assay was carried out in four replicates with negative (culture medium), positive (solution of the cytotoxic drug camptothecin at a concentration of 3 mM) controls and the solvent control (DMSO). The results of the MTT assay were evaluated on a Tecan Infinite M200 PRO (Tecan Austria GmbH, Grödig, Austria) plate spectrophotometer by comparing the optical density of a formazan solution at 570 nm in the assay and control wells. The half maximal inhibitory concentrations (IC_50_) were calculated by AAT Bioquest-calculator (https://www.aatbio.com/tools/ic50-calculator (accessed on 14 November 2022)).


*Inhibitory Activities on TRPA1 and TRPV1*


Testing for receptors ratTRPA1 (rTRPA1) and ratTRPV1 (rTRPV1), stably expressed in CHO cells, was carried out using the calcium imaging method as described in [123,124]. Fluorescence was performed using NOVOstar (BMG LABTECH, Ortenberg, Germany). The tested substances in 26% DMSO solution or 26% DMSO solution alone (negative control) (5 μL) were added to the cells loaded with the cytoplasmic calcium indicator Fluo-4AM using Fluo-4 Direct™ Calcium Assay Kit (Thermo Fisher Scientific Inc., Waltham, MA, USA) (60 μL) and incubated 5–10 min. Final DMSO concentration was 2%. Changes in cell fluorescence (λex = 485 nm, λem = 520 nm) were monitored at 25 °C before and after the addition of relevant agonist. The AITC final concentration 160 μM was used for rTRPA1 activation. For rTRPV1 activation we used (1) acid-salt buffer pH4.01 (diluted 2 times Buffer solution pH 4.01 (Mettler Toledo, Greifensee, Switzerland) containing 150 mM NaCl), (2) 2APB in 10 mM HEPES/HBSS, pH 7.4—final concentration 160 μM, or (3) 2APB in acid-salt buffer pH 4.01—final concentration 80 μM. Nonspecificity was tested on wild type CHO.

To achieve the required level of acidification, the cell medium with fluorescent dye (pH 7.4) was pre-titrated to determine the required volume of acid-salt buffer until pH 6.2 was reached. The pH level was also monitored after the test as an average of 10–12 wells. To avoid the spread of the data due to prolonged instrumental measurement, each experimental point had control in the nearest plate wells. Parameter estimations were performed with Microsoft Excel 2007 (Microsoft Corporation, Redmond, WA, USA).

#### 3.2.7. In Vivo Experiments

Laboratory animals (Sprague-Dawley rats and CD-1 mice) were obtained from the Animal Unit “Pushino” at the M.M. Shemyakin and Yu.A. Ovchinnikov from the Institute of Bioorganic Chemistry RAS, Russia. The animals were housed at natural light cycle and otherwise in a controlled environment, in propylene cages (Bioscape, Castrop-Rauxel, Germany), on standard bedding (“Zolotoi Kot”, ZKK Zolotoi Pochatok, Voronezh region, Alekasndrovka-Donskaya, Russia), supplied with feed for conventional laboratory rodents (ProKorm, “BioPro”, Novosibirsk, Russia) according to a schedule and water *ad libitum*.

Animal care and all the procedures were performed by professional staff according to Russian Federation Law №61-FZ 24 March 2010 On Circulation of Medicines, guidelines for preclinical study of medicinal products [125,126] and in accordance with approved Regulations for Care and Use of Laboratory Animals of the Research and Educational Center for Applied Chemical and Biological Research at the Perm National Research Polytechnic University.


*Acute toxicity evaluation*


The toxicity evaluation was performed on CD-1 mice. The procedure was based on OECD recommendations and guidelines for pre-clinical study of medicinal products [97,125]. The tested compounds in 1% starch mucilage were injected intraperitoneally into three mice. Then, animals were observed during 14 days, the number of deaths was counted and % of viability calculated.


*The hot plate test*


The hot plate test was conducted according to established guidelines [125,126] on Sprague-Dawley (SD) rats (3 male and 3 female rodents per group). The compounds were intraperitoneally administered in the form of suspensions in 1% starch mucilage. Negative control group animals received vehicle only (1% starch mucilage). Diclofenac sodium (Hemofarm, Vršac, Serbia, 10 mg/kg in 1% starch mucilage solution) or metamizole sodium (PAO Biosynthesis, Sun Pharma, Penza, Russia, 15 mg/kg in 1% starch mucilage solution) were used as reference drugs. One and two hours after the injection, rats were placed on an electrically heated to 50 °C plate (Hot plate 60,200 series, TSE-systems, Bad Homburg, Germany) in a Plexiglas cylindrical restrainer (19 cm diameter × 30 cm). The nociceptive response time was measured by observing the appearance of rats’ movements (e.g., jumping, hind paw licking or shaking). Maximal cutoff time was set as 30 s regardless of the response in order to preclude skin damage.

Differences between the values of the latency period for the treated (experimental) and control groups of animals were considered significant at *p* ≤ 0.05. Analgesic activity was expressed as an increase (in percent) in the response time to nociceptive stimulation in the group of animals that received the substance, compared with the control group of animals, and was calculated only for those substances that significantly increased the latency period, according to the formula: AA = ((t_tr_ − t_c_) × 100)/t_c_, where t_tr_ is the response time to nociceptive stimulation in the group of animals that received the substance or reference drug; t_c_ is the response time to nociceptive stimulation in animals of the control group.


*The carrageenan-induced paw edema model.*


Anti-inflammatory activity was studied in Sprague-Dawley rats (3 male and 3 female animals per group) by using the common carrageenan-induced paw edema model [125,126]. The investigated compounds in 1% starch mucilage were intraperitoneally administered at the corresponding dose to a treated group of rodents 30 min before the injection of 0.1 mL of freshly prepared 1% carrageenan (λ-carrageenan, type IV; Sigma Aldrich, an affiliate of Merck KGaA, Darmstadt, Germany) solution in the right hind paw plantar surface. The negative control group was treated with 1% starch mucilage solution only. Diclofenac sodium (10 mg/kg in 1% starch mucilage solution) was used as a reference drug. Paw volumes were measured oncometrically with water plethysmometer (TSE Volume Meter, Bad Homburg, Germany) before carrageenan administration (“zero time”) and at 1st, 3rd, 5th and 24th h after its injection. An anti-inflammatory activity of the sample at a certain time point was calculated in two steps. First, the percentage of an edema increment (i.e., relative volume change for the inflamed paw vs. “zero time” value) both in the negative control group (P_c_) and in the treated group (P_tr_) were calculated for this time point for each animal. The difference between treated and control group was considered significant at *p* < 0.05. Then, for such cases, swelling inhibition was calculated according to the formula: 100% (P_c_ − P_tr_)/P_c_ and considered as a measure of anti-inflammatory activity of the sample.

### 3.3. Quantum-Mechanical Calculations

#### 3.3.1. Ligand Preparation

The optimization of the geometric parameters of the ligand followed by the solution of the frequency task was carried out using the most popular DFT calculation method B3LYP [127] with the dispersion correction D2 [128] and the 6–31G** basis set [129]. To estimate the acidity constant, the calculations were carried out in the gas phase approximation and taking into account the solvent (PCM model, water) [130].

Transient receptor potential cation channel subfamily V member 1 (TrpV1) was considered as potential biological target. Crystallographic data of the complex receptor with the reference ligand capsaicin (PDB [131] code 7LR0 [132]) were downloaded from non-commercial Protein data bank. The protein is a transmembrane homotetramer containing 4 intermembrane capsaicin binding pockets. To simplify the calculation model, we considered chains A and B with the corresponding vanilloid capsaicin binding site formed by them. The geometrical parameters of the protein were prepared for the calculation appropriately: hydrogen atoms were added and minimized, bond multiplicities and side chains of amino acids were restored, water molecules and excess low molecular weight compounds were removed. The system was optimized by the force field method OPLS3e [133].

#### 3.3.2. Molecular Docking

Molecular docking was performed using an induced fit docking (or flexible) protocol under the following conditions: flexible protein and ligand, grid size 15Å, a.a. within a radius of 5Å additionally optimized taking into account the effect of the ligand on the side chains. Twenty docking positions were set. Docking results were ranked based on the evaluation of the following scoring functions: docking score is the main score of docking results, consisting of various energy terms, taking into account clash interaction penalties, e-model is the energy of ligand clustering in the binding site, and IFD score is the energy of the ligand-protein complex taking into account the term of the Coulomb interaction.

## 4. Conclusions

Summarizing the data obtained, we can conclude the prospect of 4-AHP backbone usage for the design of bioactive compounds. The presence of a polyfluoroalkyl substituent in their structure realized an anticarboxylesterase activity, which was removed by the introduction of sulfonyl groups or a bromine atom. However, the variation in the structural fragments of the 4-AHP core can be used to develop selective CES inhibitors based on this scaffold. The presence of an NH fragment in the molecule, as well as the replacement of a polyfluoroalkyl residue by a methyl group, opens the ability to bind free radicals. Admittedly, the studied compounds fail in antifungal activity, but their high anti-gonorrhea activity may convert further to antibacterial agent creation. The antimicrobial effect of the pyrazolones can be significantly enhanced by the introduction of some target pharmacophore fragments.

It is very significant that the compounds have no toxicity on rats at a dose of 300 mg/kg. However, CF_3_-derivatives having an unsubstituted nitrogen atom exhibited a moderate cytotoxicity to normal human fibroblasts, which was removed by the introduction of a phenyl substituent or decreased by elongation of the polyfluoroalkyl residue. Note that sulfonyl-containing pyrazolones are not cytotoxic. Definitely, the 4-AHP skeleton could be useful in the development of anticancer agents, and in this aspect the presence of the NH moiety has proved to be important.

The 4-AHPs have special prospects for the development of effective analgesics. In this regard, the introduction of the methylsulfonyl groups proved to be useful. This not only enhanced the antinociceptive effect, but also reduced the cytotoxicity and anticarboxylesterase activity that could exclude possible undesirable effects, namely, drug–drug interactions. Among all the compounds, pyrazolones **5m** and **5r** are obviously leading compounds for further in-depth study. It is also interesting that compound **5m** showed an anti-inflammatory effect in contrast to other tested pyrazolones without sulfonyl groups, which may indicate its inhibitory action on cyclooxygenases. For analgesia active pyrazolones without sulfonyl substituents, we suggested a mechanism of action through blocking the TRPV1 receptor. It was confirmed by molecular docking and in vitro experimental studies for pyrazolone **6b**. In general, the direction of the search for antinociceptive agents in the series of 4-AHPs, including polyfluoroalkyl-containing ones, seems very promising for further development.

## Data Availability

Not applicable.

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
