# Peer review of "Powerful Potential of Polyfluoroalkyl-Containing 4-Arylhydrazinylidenepyrazol-3-ones for Pharmaceuticals"

_molecules, 2022, doi:10.3390/molecules28010059_

Round 1

Reviewer 1 Report

Saloutin and coworkers have reported the synthesis of fluorinated 4-arylhydrazinylidenepyrazol-3-ones and their biological evaluation. All the compounds were properly characterized (with exception of the reported ones). A number of in-vivo and in-vitro biological studies were performed. 

I personally enjoyed reading the manuscript and will be happy to recommend the publication of the manuscript with the following corrections/modifications:

1. Scheme 1, it will be ideal to dray the chemical structure of all the synthesized compounds with their respective yields. Besides, add the reaction conditions and stoichiometric quantity of the substrates.

2. Add HNMR and CNMR data in the manuscript for the reported compounds also. 

Author Response

We appreciate Reviewer's time considering the paper and are grateful for  efforts and for the positive evaluation of our manuscript.

Reviewer form:

Saloutin and coworkers have reported the synthesis of fluorinated 4-arylhydrazinylidenepyrazol-3-ones and their biological evaluation. All the compounds were properly characterized (with exception of the reported ones). A number of in-vivo and in-vitro biological studies were performed. 

I personally enjoyed reading the manuscript and will be happy to recommend the publication of the manuscript with the following corrections/modifications:

  1. Scheme 1, it will be ideal to dray the chemical structure of all the synthesized compounds with their respective yields. Besides, add the reaction conditions and stoichiometric quantity of the substrates.

A: We changed the Scheme 1 and added the Table 1 with substituents and yields.

  1. Add HNMR and CNMR data in the manuscript for the reported compounds also. 

A: All compounds have described using the NMR spectra (please, see Experimental part and discussion on the page 7). In addition, the copies of NMR spectra are in Supplementary Material

Reviewer 2 Report

The described studies cover a very large group of compounds.

They concern the method of synthesis (three), the likely directions of biological activity and molecular targets, and a possible mechanism of action has been proposed.

The description is extensive, the research methods are correct. There are several questions and comments regarding study/test selection and manuscript preparation.

The text contains abbreviations related to biological research, the development of which can be found later in the text or in the description of the research. Abbreviations should be explained the first time they occur in the text for use in subsequent lines, e.g. lines 71, 351, 357, 358 and explanatory notes 80, 361, 363, respectively.

I propose to edit the text of the introduction so as to explain the abbreviations and discuss the desirability of performing selected tests.

Editor's note: Scheme 1 describes the structure of many compounds, the expansion of the symbols RF/R1 is very crowded and not very clear - consider the ordering of the substituents in the table.

Line 419 - "According to WOS" please explain.

Hot plate test - results were compared in a table with the activity of reference compounds, what prompted you to choose diclofenac as a reference in this study? A reasonable choice is a recognized drug with a similar structure, Metamizole. However, the effects of diclofenac have not been comprehensively discussed in comparison to the compounds studied. The experiment was carried out in one portion only (15mg/kg). Of course, this allows you to compare their effects at this dose, but the result has not been discussed in relation to the reference substance. The hot plate test approximates central antinociceptive properties. What made you choose diclofenac? There are preliminary studies that are considering supplementing the study with sequential dilutions of the lead compounds and/or the formalin test, a two-phase test that more accurately describes the properties of the substance and the effect on neuropathic pain.

Acute toxicity. Full safety at a dose of 300mg/kg does not mean no acute toxicity.

I propose to refer the obtained results to the acute toxicity of reference compounds. Anti-inflammatory effect. If the molecular target is cyclooxygenase, the choice of diclofenac is debatable.

The classical reference compound would be meloxicam or celecoxib.

Analytical data. For some of the final compounds (line 943, derivative 9h), differences in the content of atoms greater than +/- 0.4% from the calculated amounts are noticeable (C;S elemental analysis). Has a mass spectroscopy study of the 9h derivative been performed?

The dependence of the action on the structure of the described derivative groups has been discussed in detail. The summary of the QSARs could be included after grouping due to modifications of the structure of the final compounds towards specific molecular targets - antibacterial, analgesic, anti-inflammatory in the final conclusions.

Author Response

We appreciate Reviewer's time considering the paper and are grateful for  efforts and for the positive evaluation of our manuscript.

Reviewer form:

The described studies cover a very large group of compounds.

They concern the method of synthesis (three), the likely directions of biological activity and molecular targets, and a possible mechanism of action has been proposed.

The description is extensive, the research methods are correct. There are several questions and comments regarding study/test selection and manuscript preparation.

The text contains abbreviations related to biological research, the development of which can be found later in the text or in the description of the research. Abbreviations should be explained the first time they occur in the text for use in subsequent lines, e.g. lines 71, 351, 357, 358 and explanatory notes 80, 361, 363, respectively.

I propose to edit the text of the introduction so as to explain the abbreviations and discuss the desirability of performing selected tests.

 It was corrected, but some abbreviations (e.g., HIV, FDA) are commonly used. The other abbreviations (e.g., antioxidant tests) were explained when they were used at first in the text. If all explanation of abbreviations is in the introduction, it will be too large. We would be grateful if the introduction part remains in the current form.

Editor's note: Scheme 1 describes the structure of many compounds, the expansion of the symbols RF/R1 is very crowded and not very clear - consider the ordering of the substituents in the table.

We changed the Scheme 1 and added the Table 1 with substituents and yields.

Line 419 - "According to WOS" please explain.

Edited. It should be WHO.

Hot plate test - results were compared in a table with the activity of reference compounds, what prompted you to choose diclofenac as a reference in this study? A reasonable choice is a recognized drug with a similar structure, Metamizole. However, the effects of diclofenac have not been comprehensively discussed in comparison to the compounds studied. The experiment was carried out in one portion only (15mg/kg). Of course, this allows you to compare their effects at this dose, but the result has not been discussed in relation to the reference substance. The hot plate test approximates central antinociceptive properties. What made you choose diclofenac?

Diclofenac is often used as a reference drug for evaluation of the analgesic effect of substances because it is widely applied as an analgesic. In addition, this drug was used to compare acute toxicity (data has been added to Table 8)

There are preliminary studies that are considering supplementing the study with sequential dilutions of the lead compounds and/or the formalin test, a two-phase test that more accurately describes the properties of the substance and the effect on neuropathic pain

In this work, we performed a large number of biological tests to identify the most probable activity of the synthesized compounds. At the same time, some results are of particular interest for the further research. For example, there are inspiring data on antinociceptive activity. Herein, we used the generally accepted Hot Plate test to identify the analgesic effect of compounds. Undoubtedly, we will continue our research in this promising area. Further research will include expanding the range of applied models for assessing the analgesic effect, including the formalin test. All these data deserve another targeted publication.

Acute toxicity. Full safety at a dose of 300mg/kg does not mean no acute toxicity. I propose to refer the obtained results to the acute toxicity of reference compounds.

The article did not contain the statement that substances cannot be toxic at a dose higher than 300 mg/kg. The evaluation of acute toxicity at this value was used to select the dose for primary screening in the selected model for assessing antinociceptive effects. The dose of the test substance in an in vivo model should not exceed 0.1xLD50. In the absence of acute toxicity at a dose of 300 mg/kg, this condition is reliably met for a dose of the test substance of 15 mg/kg in a model experiment.

I propose to refer the obtained results to the acute toxicity of reference compounds.

Data on acute toxicity are involved in Table 8.

Anti-inflammatory effect. If the molecular target is cyclooxygenase, the choice of diclofenac is debatable. The classical reference compound would be meloxicam or celecoxib.

Diclofenac is widely used as the reference drug for evaluation of anti-inflammatory action of compounds [e.g, M.S. El-Shoukrofy, H.A. Abd El Razik, O.M. AboulWafa, A.E. Bayad, I.M. El-Ashmawy, Pyrazoles containing thiophene, thienopyrimidine and thienotriazolopyrimidine as COX-2 selective inhibitors: Design, synthesis, in vivo anti-inflammatory activity, docking and in silico chemo-informatic studies, Bioorg. Chem. 85 (2019) 541–557. doi:10.1016/j.bioorg.2019.02.036; R. Surendra Kumar, I.A. Arif, A. Ahamed, A. Idhayadhulla, Anti-inflammatory and antimicrobial activities of novel pyrazole analogues, Saudi J. Biol. Sci. 23 (2016) 614–620. doi:10.1016/j.sjbs.2015.07.005. M. Ahmed, M.A. Qadir, A. Hameed, M. Imran, M. Muddassar, Screening of curcumin-derived isoxazole, pyrazoles, and pyrimidines for their anti-inflammatory, antinociceptive, and cyclooxygenase-2 inhibition, Chem. Biol. Drug Des. 91 (2018) 338–343. doi:10.1111/cbdd.13076; S.S. Abd El-Karim, H.S. Mohamed, M.F. Abdelhameed, A. El-Galil E. Amr, A.A. Almehizia, E.S. Nossier, Design, synthesis and molecular docking of new pyrazole-thiazolidinones as potent anti-inflammatory and analgesic agents with TNF-α inhibitory activity, Bioorg. Chem. 111 (2021) 104827. doi:10.1016/j.bioorg.2021.104827]. We chose diclofenac in our work in the similar manner. Meloxicam or celecoxib would be used as reference drugs in the evaluation of compounds as the selective inhibitors of cyclooxygenase-2. In this work, we did not set such task.

Analytical data. For some of the final compounds (line 943, derivative 9h), differences in the content of atoms greater than +/- 0.4% from the calculated amounts are noticeable (C;S elemental analysis). Has a mass spectroscopy study of the 9h derivative been performed?

It is inaccuracy for compound 6h (not 9h). We checked compounds with S element again and added elemental analysis data from another equipment. Note that the NMR spectra of compound 6h was appropriate.

 The dependence of the action on the structure of the described derivative groups has been discussed in detail. The summary of the QSARs could be included after grouping due to modifications of the structure of the final compounds towards specific molecular targets - antibacterial, analgesic, anti-inflammatory in the final conclusions.

According to Molecules template, Conclusion is at the end of paper (on the pages 35-36), where contain QSARs for the structure of the final compounds towards antibacterial, antioxidant, cytotoxic, analgesic, anti-inflammatory activities.

Round 2

Reviewer 2 Report

The answers and explanations are sufficient. I am still of the opinion that diclofenac is not the most accurate reference agent in the hot plate test. I recommend the publication of the manuscript.